# Diverse ancestry whole-genome sequencing association study identifies *TBX5* and *PTK7* as susceptibility genes for posterior urethral valves

Melanie MY Chan[1], Omid Sadeghi-Alavijeh[1], Filipa M Lopes[2], Alina C Hilger[3,4], Horia C Stanescu[1], Catalin D Voinescu[1], Glenda M Beaman[5,6], William G Newman[5,6], Marcin Zaniew[7], Stefanie Weber[8], Yee Mang Ho[2], John O Connolly[1,9], Dan Wood[9], Carlo Maj[10,11], Alexander Stuckey[12], Athanasios Kousathanas[12], Genomics England Research Consortium, Robert Kleta[1,13], Adrian S Woolf[2,14], Detlef Bockenhauer[1,13], Adam P Levine[1,15], Daniel P Gale[1]*

[1]Department of Renal Medicine, University College London, London, United Kingdom; [2]Division of Cell Matrix Biology & Regenerative Medicine, School of Biological Sciences, Faculty of Biology, Medicine and Health, University of Manchester, Manchester, United Kingdom; [3]Children's Hospital, University of Bonn, Bonn, Germany; [4]Institute of Human Genetics, University of Bonn, Bonn, Germany; [5]Manchester Centre for Genomic Medicine, Manchester University NHS Foundation Trust, Manchester, United Kingdom; [6]Evolution and Genomic Sciences, School of Biological Sciences, University of Manchester, Manchester, United Kingdom; [7]Department of Pediatrics, University of Zielona Góra, Zielona Gora, Poland; [8]Department of Pediatric Nephrology, University of Marburg, Marburg, Germany; [9]Department of Adolescent Urology, University College London Hospitals NHS Foundation Trust, London, United Kingdom; [10]Center for Human Genetics, University of Marburg, Marburg, Germany; [11]Institute for Genomic Statistics and Bioinformatics, Medical Faculty, University of Bonn, Bonn, Germany; [12]Genomics England, Queen Mary University of London, London, United Kingdom; [13]Nephrology Department, Great Ormond Street Hospital for Children NHS Foundation Trust, London, United Kingdom; [14]Royal Manchester Children's Hospital, Manchester University NHS Foundation Trust, Manchester Academic Health Science Centre, Manchester, United Kingdom; [15]Research Department of Pathology, University College London, London, United Kingdom

*For correspondence: d.gale@ucl.ac.uk

Group author details: Genomics England Research Consortium See page 23

**Abstract** Posterior urethral valves (PUV) are the commonest cause of end-stage renal disease in children, but the genetic architecture of this rare disorder remains unknown. We performed a sequencing-based genome-wide association study (seqGWAS) in 132 unrelated male PUV cases and 23,727 controls of diverse ancestry, identifying statistically significant associations with common variants at 12q24.21 (p=7.8 × 10$^{-12}$; OR 0.4) and rare variants at 6p21.1 (p=2.0 × 10$^{-8}$; OR 7.2), that were replicated in an independent European cohort of 395 cases and 4151 controls. Fine mapping and functional genomic data mapped these loci to the transcription factor *TBX5* and planar cell polarity gene *PTK7*, respectively, the encoded proteins of which were detected in the developing urinary tract of human embryos. We also observed enrichment of rare structural variation intersecting with

candidate *cis*-regulatory elements, particularly inversions predicted to affect chromatin looping (p=3.1 × 10$^{-5}$). These findings represent the first robust genetic associations of PUV, providing novel insights into the underlying biology of this poorly understood disorder and demonstrate how a diverse ancestry seqGWAS can be used for disease locus discovery in a rare disease.

## Editor's evaluation

Prior work has linked posterior urethral valves (PUV), a common cause of end stage renal disease in children, with chromosomal abnormalities and rare copy number variants, but the genetic causes of PUV remain incompletely defined. In this study, the authors have used a diverse ancestry whole-genome sequencing association study to identify two novel genes, *TBX5* and *PTK7*, and an enrichment of rare duplications and inversions affecting candidate *cis*-regulatory elements as possible causes of this condition. These findings represent the first robust genetic associations of PUV, provide the foundation for developing a mechanistic understanding of the cause of this condition, and demonstrate how a diverse ancestry seqGWAS can be used for disease locus discovery in a rare disease.

## Introduction

Posterior urethral valves (PUV) are the commonest cause of end-stage renal disease (ESRD) in children, affecting 1 in 4000 male births (*Brownlee et al., 2019*; *Thakkar et al., 2015*) and result in congenital bladder outflow obstruction. It is an exclusively male disorder, with over a third of those affected developing ESRD (i.e., requirement for dialysis or kidney transplantation) before the age of 30 years (*Heikkilä et al., 2011*; *Sanna-Cherchi et al., 2009*) and often associated with renal dysplasia, vesicoureteral reflux, and bladder dysfunction which are poor prognostic factors for renal survival (*Sanna-Cherchi et al., 2009*). Management involves endoscopic valve ablation to relieve the obstruction, however the majority of affected children have long-term sequelae related to ongoing bladder dysfunction (*DeFoor et al., 2008*).

The pathogenesis of PUV is poorly understood (*Krishnan et al., 2006*). Although usually sporadic, familial clustering and twin studies suggest an underlying genetic component, although Mendelian inheritance is rare (*Chiaramonte et al., 2016*; *Frese et al., 2019*; *Morini et al., 2002*; *Schreuder et al., 2008*; *Weber et al., 2005*). Several monogenic causes of other congenital bladder outflow obstruction disorders have been described including *BNC2* in urethral stenosis (*Kolvenbach et al., 2019*); *FLNA* in syndromic urethral anomalies (*Wade et al., 2021*); *HPSE2* and *LRIG2* in urofacial syndrome (*Daly et al., 2010*; *Stuart et al., 2008*); *CHRM3* in prune-belly like syndrome (*Weber et al., 2011*); and *MYOCD* in megabladder (*Houweling et al., 2011*). However, a monogenic etiology for isolated PUV has not been identified. Case reports of chromosomal abnormalities resulting in PUV as part of a wider syndrome (*Demirkan, 2021*; *Houcinat et al., 2011*; *Tong et al., 2015*) and microarray-based studies linking rare copy number variants (CNVs) with PUV (*Boghossian et al., 2016*; *Caruana et al., 2016*; *Faure et al., 2016*; *Schierbaum et al., 2021*; *Verbitsky et al., 2019*) suggest that structural variation, and in particular duplications (*Schierbaum et al., 2021*; *Verbitsky et al., 2019*), may be important, but no recurrent CNVs have demonstrated consistent association with PUV. The observation that isolated PUV does not usually follow a classical Mendelian inheritance pattern, and that a monogenic cause has not been identified, suggests that the underlying genetic architecture of this rare disorder is likely to be complex.

Here, we used whole-genome sequencing (WGS) data from a diverse ancestry cohort to perform a sequencing-based genome-wide association study (seqGWAS) investigating how common, low-frequency, and rare single-nucleotide and structural variation contribute to this complex disorder. The recently proposed term seqGWAS is used to distinguish this approach from the well-established genome-wide association study (GWAS) design based on sampling of common variation across the genome using high-throughput single-nucleotide polymorphism genotyping with or without imputation (*McMahon et al., 2021*). Our seqGWAS identified significant associations with common and rare variation that implicated *TBX5* (T-Box Transcription Factor 5) and *PTK7* (Protein Tyrosine Kinase 7), respectively, as well as enrichment of structural variation affecting candidate *cis*-regulatory elements

(cCREs). In addition, we demonstrate that a well-controlled diverse ancestry sequencing-based GWAS can increase power for disease locus discovery and facilitate the fine mapping of causal variants.

## Results

We analysed WGS data from 132 unrelated male probands with PUV and 23,727 non-PUV controls (unaffected relatives without known kidney disease), recruited to the UK's 100,000 Genomes Project (100KGP) (*Smedley et al., 2021*; see *Figure 1* for study workflow). The available dataset (version 10) combined WGS data, clinical phenotypes standardized using Human Phenotype Ontology (HPO) codes, and comprehensive hospital clinical records for 89,139 individuals with cancer, rare disease, and their unaffected relatives. None of the cases included had received a definitive genetic diagnosis through the clinical arm of the 100KGP. Two individuals had a pathogenic and likely pathogenic variant affecting *HNF1B* and *FOXC1*, respectively, but these were not considered causal for PUV (see Appendix 1). Given the small number of recruited cases with this rare disorder, we chose to jointly analyse individuals from diverse ancestral backgrounds, thereby preserving sample size and boosting power. To mitigate confounding due to population structure whilst using this mixed-ancestry approach, we performed ancestry-matching of cases and controls using weighted principal components (*Figure 1—figure supplement 1*) and utilized SAIGE which employs a scalable generalized logistic mixed model (GLMM) to account for relatedness between individuals and a saddlepoint approximation to avoid inflated type I error seen with case-control imbalance (*Zhou et al., 2016a*). Clinical characteristics and genetic ancestry of the cases and controls are detailed in *Table 1*.

### Variation at 12q24.21 and 6p21.1 is associated with PUV

To determine the contribution of common and low-frequency variation to PUV, we carried out a seqGWAS using 19,651,224 single-nucleotide variants (SNVs) and indels with minor allele frequency (MAF) ≥0.1%. Statistically significant ($p < 5 \times 10^{-8}$) association was detected at two loci (*Figure 2* and *Table 2*). The genomic inflation factor ($\lambda$) of 1.04 confirmed population stratification was well controlled in this diverse ancestry cohort, although this may represent an underestimate given the low power of the cohort (*Figure 2—figure supplement 1*).

At 12q24.21, the lead intergenic variant (rs10774740) was common (MAF 0.37) and reached $p=7.81 \times 10^{-12}$ (OR 0.40; 95% CI 0.31–0.52; *Figure 3*). A rare (MAF 0.007) variant (rs144171242) at 6p21.1, located in an intron of *PTK7*, was also significant at $p=2.02 \times 10^{-8}$ (OR 7.20; 95% CI 4.08–12.70; *Figure 4*). The meanDP (mean sequencing depth) and meanGQ (mean genotype quality) at these sites were 33.43 and 133.28, for rs10774740 and 29.34 and 75.59 for rs144171242, respectively, confirming these variants were high-quality calls. Conditional analysis did not identify secondary independent signals at either locus and epistasis was not detected between the two lead variants (p=0.10).

Due to the male nature of PUV, we also conducted a sex-specific analysis using only male controls (n=10,425) to determine whether any additional signals could be detected by removing females with potentially undetected genitourinary phenotypes. Both lead variants showed stronger evidence of association: rs10774740 at 12q24.21 ($p=7.08 \times 10^{-12}$; OR 0.40; 95% CI 0.31–0.52) and rs144171242 at 6p21.1 ($p=1.79 \times 10^{-8}$; OR 7.40; 95% CI 4.14–13.22). No significant associations on chromosome X were identified.

Gene and gene set analysis was carried out to assess the joint effect of common and low-frequency variants and identify potential functional pathways associated with PUV, however, no genes (*Figure 2—source data 1*) or pathways (*Figure 2—source data 2*) reached statistical significance after correction for multiple testing.

### 12q24.21 and 6p21.1 replicate in an independent cohort

We next carried out a replication study in an independent European cohort consisting of 395 individuals with PUV: 333 from Poland and Germany, recruited through the CaRE for LUTO (Cause and Risk Evaluation for Lower Urinary Tract Obstruction) Study, and 62 from the UK; 4151 male individuals recruited to the cancer arm of the 100KGP were used as controls. The ancestry of a subset of cases (n=204) for whom genome-wide genotyping data was available and the control cohort (n=4151) was confirmed as European (*Figure 5*). The UK PUV patients and the 100KGP cancer control cohort had not been included in the discovery analyses.

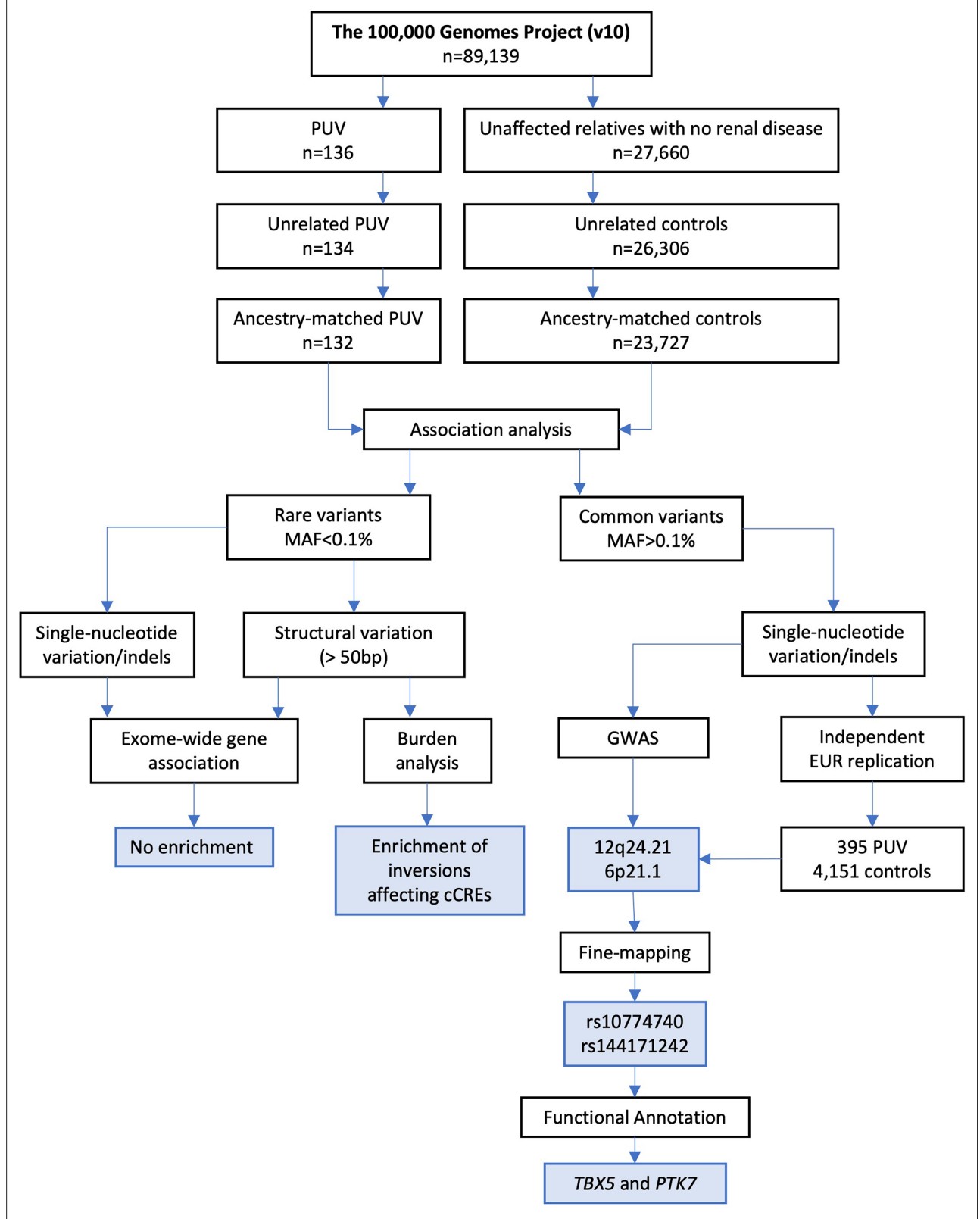

**Figure 1.** Study workflow. The flowchart shows the number of samples included at each stage of filtering, the analytical strategies employed, and the main findings (blue boxes). PUV, posterior urethral valves; MAF, minor allele frequency; GWAS, genome-wide association study; EUR, European; cCRE, candidate *cis*-regulatory element.

*Figure 1 continued on next page*

*Figure 1 continued*

The online version of this article includes the following figure supplement(s) for figure 1:

**Figure supplement 1.** Principal component analysis (PCA) showing the first eight principal components for matched cases (blue) and controls (black) and unmatched cases (orange) and controls (grey).

The lead variants at the top four loci with p<5 × 10$^{-7}$ were tested for replication. Association at both genome-wide significant lead variants was replicated although with smaller effect sizes (*Table 2*): rs10774740 (p=5.17 × 10$^{-3}$; OR 0.80; 95% CI 0.68–0.93) and rs144171242 (p=7.21 × 10$^{-3}$; OR 2.18; 95% CI 1.22–3.90). Two further loci with suggestive evidence of association (10q11.2; rs1471950716; p=1.45 × 10$^{-7}$ and 14q21.1; rs199975325; p=2.52 × 10$^{-7}$) did not replicate (*Table 2—source data 1*).

To confirm this signal was not being driven by case-control imbalance, we repeated the analysis with 500 male controls (a case:control ratio of 1:1.3). rs10774740 at the 12q24.21 locus remained significant (p=9.9 × 10$^{-3}$; OR 0.77; 95% CI 0.63–0.94). rs144171242 at the 6p21.1 locus did not reach statistical significance, but this is likely due to insufficient power (p=0.06; OR 2.24; 95% CI 0.93–5.36).

**Table 1.** Clinical characteristics and genetic ancestry of the discovery cohort.

| | | PUV (n=132) | Controls (n=23,727) |
|---|---|---|---|
| Median age (range) | | 13 (2–66) | |
| Males (%) | | 132 (100) | 10,425 (43.9) |
| PCA determined ancestry | | | |
| | EUR (%) | 89 (67.4) | 19,418 (81.8) |
| | SAS (%) | 18 (13.6) | 2847 (12.0) |
| | AFR (%) | 11 (8.3) | 449 (1.9) |
| | AMR (%) | 0 (0) | 7 (0.03) |
| | Admixed (%) | 14 (10.6) | 1006 (4.2) |
| Additional renal/urinary phenotypes | | | |
| | Hydronephrosis (%) | 56 (42.4) | |
| | Bladder abnormality (%) | 32 (24.2) | |
| | Hydroureter (%) | 30 (22.7) | |
| | VUR (%) | 27 (20.5) | |
| | Renal dysplasia (%) | 16 (12.1) | |
| | Hypertension (%) | 11 (8.3) | |
| | Renal agenesis (%) | 8 (6.1) | |
| | Recurrent UTIs (%) | 5 (3.8) | |
| | Renal hypoplasia (%) | 4 (3.0) | |
| | Renal duplication (%) | 2 (1.5) | |
| Extrarenal manifestations (%) | | 35 (26.5) | |
| | Cardiac anomaly (%) | 4 (3.0) | |
| | Neurodevelopmental disorder (%) | 7 (5.3) | |
| Family history (%) | | 5 (3.8) | |
| End-stage renal disease (%) | | 23 (17.4) | |
| Median age ESRD (range) | | 14 (0–39) | |

PUV, posterior urethral valves; PCA, principal component analysis; EUR, European; SAS, South Asian; AFR, African; AMR, Latino/Admixed American; VUR, vesico-ureteral reflux; UTI, urinary tract infection; ESRD, end-stage renal disease.

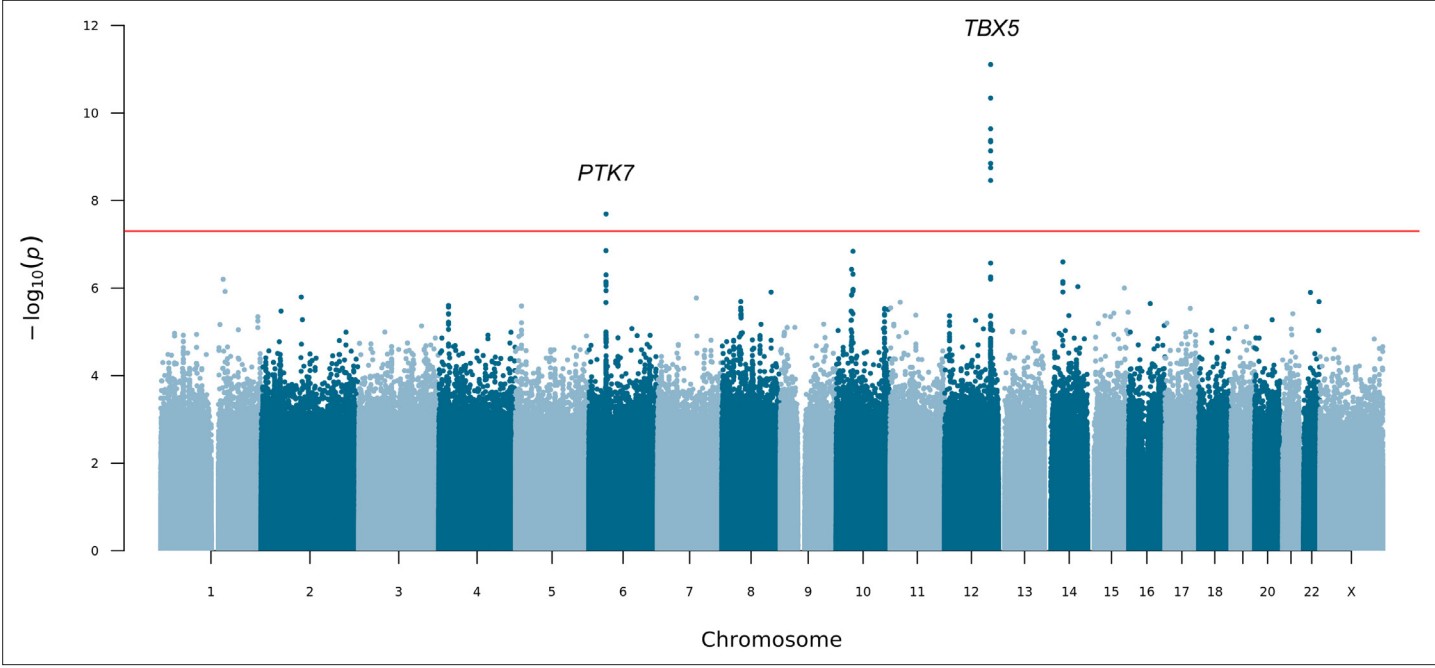

**Figure 2.** Manhattan plot for mixed-ancestry sequencing-based genome-wide association study (seqGWAS). A genome-wide single-variant association study was carried out in 132 unrelated posterior urethral valves (PUV) cases and 23,727 controls for 19,651,224 variants with minor allele frequency (MAF) >0.1%. Chromosomal position (GRCh38) is denoted along the x axis and strength of association using a $-\log_{10}(p)$ scale on the y axis. Each dot represents a variant. The red line indicates the Bonferroni adjusted threshold for genome-wide significance ($p<5 \times 10^{-8}$). The gene in closest proximity to the lead variant at significant loci is listed.

The online version of this article includes the following source data and figure supplement(s) for figure 2:

**Source data 1.** Joint analysis of common and low-frequency variation (minor allele frequency [MAF] >0.1%) by gene across the exome.

**Source data 2.** Joint analysis of common and low-frequency variation (minor allele frequency [MAF] >0.1%) by gene set.

**Figure supplement 1.** Quantile-quantile (Q–Q) plot for the mixed-ancestry genome-wide association study (GWAS) displaying the observed vs. the expected $-\log_{10}(p)$ for each variant tested.

**Figure supplement 2.** Power calculations for the mixed-ancestry genome-wide association study (GWAS) were performed at various minor allele frequencies (MAF) using 132 cases and 23,727 controls under an additive genetic model to achieve genome-wide significance of $p<5 \times 10^{-8}$.

For a rare variant such as rs144171242 (MAF 0.007), a replication study with 500 controls is only powered to detect association with variants that have a large effect size (OR >3.5).

## Diverse ancestry analysis increases power for discovery

To ascertain whether the observed associations were being driven by a specific ancestry group, we next repeated the seqGWAS using a subgroup of genetically defined European individuals (88 cases and 17,993 controls) and 16,938,500 variants with MAF ≥0.1%. The 12q24.21 locus remained genome-wide significant (*Figure 6* and *Figure 6—figure supplement 1*), however the lead variant (rs2555009) in the region showed weaker association ($p=4.02 \times 10^{-8}$; OR 0.43; 95% CI 0.12–0.73) than rs10774740, the lead variant in the diverse ancestry analysis (*Figure 6—source data 1*). Interestingly, the two variants were not in strong linkage disequilibrium (EUR LD; $r^2=0.55$). The lead variant at 6p21.1 from the diverse ancestry analysis did not reach genome-wide significance in the European-only study (rs144171242; $p=3.60 \times 10^{-5}$; OR 5.90; 95% CI 2.88–12.11) suggesting either that this signal may be being driven partly by non-Europeans or the result of loss of power due to reduced sample size. p-Values and effect sizes were strongly correlated between the diverse ancestry and European-only seqGWAS (*Figure 6—figure supplement 2*), demonstrating that inclusion of individuals from diverse backgrounds to increase sample size can be an effective way to boost power and discover new disease loci, even in a small cohort.

As the numbers of African, South Asian, and admixed ancestry individuals were too small to reliably carry out subgroup association analyses and subsequent meta-analysis, we instead compared

**Table 2.** Association statistics for significant genome-wide loci.

The lead variant with the lowest p-value at each locus is shown with genome-wide significance defined as p<5 × 10$^{-8}$. Genomic positions are with reference to GRCh38. Discovery p-values were derived using SAIGE generalized logistic mixed model association testing and replication p-values using a two-sided Cochran Armitage Trend Test. CHR, chromosome; POS, position; OR, odds ratio; CI, confidence interval; EAF, effect allele frequency.

| Lead variant | CHR:POS | Effect Allele | Closest gene | P value Discovery | P value Replication | OR (95% CI) Discovery | OR (95% CI) Replication | Case EAF Discovery | Case EAF Replication | Control EAF Discovery | Control EAF Replication |
|---|---|---|---|---|---|---|---|---|---|---|---|
| rs10774740 | chr12:114228397 | T | TBX5 | 7.81×10$^{-12}$ | 5.17×10$^{-3}$ | 0.4 (0.31–0.52) | 0.8 (0.68–0.93) | 0.19 | 0.31 | 0.37 | 0.36 |
| rs144171242 | chr6:43120356 | G | PTK7 | 2.02×10$^{-8}$ | 7.21×10$^{-3}$ | 7.2 (4.08–12.70) | 2.18 (1.22–3.90) | 0.05 | 0.018 | 0.01 | 0.008 |

The online version of this article includes the following source data for table 2:

**Source data 1.** Replication study.

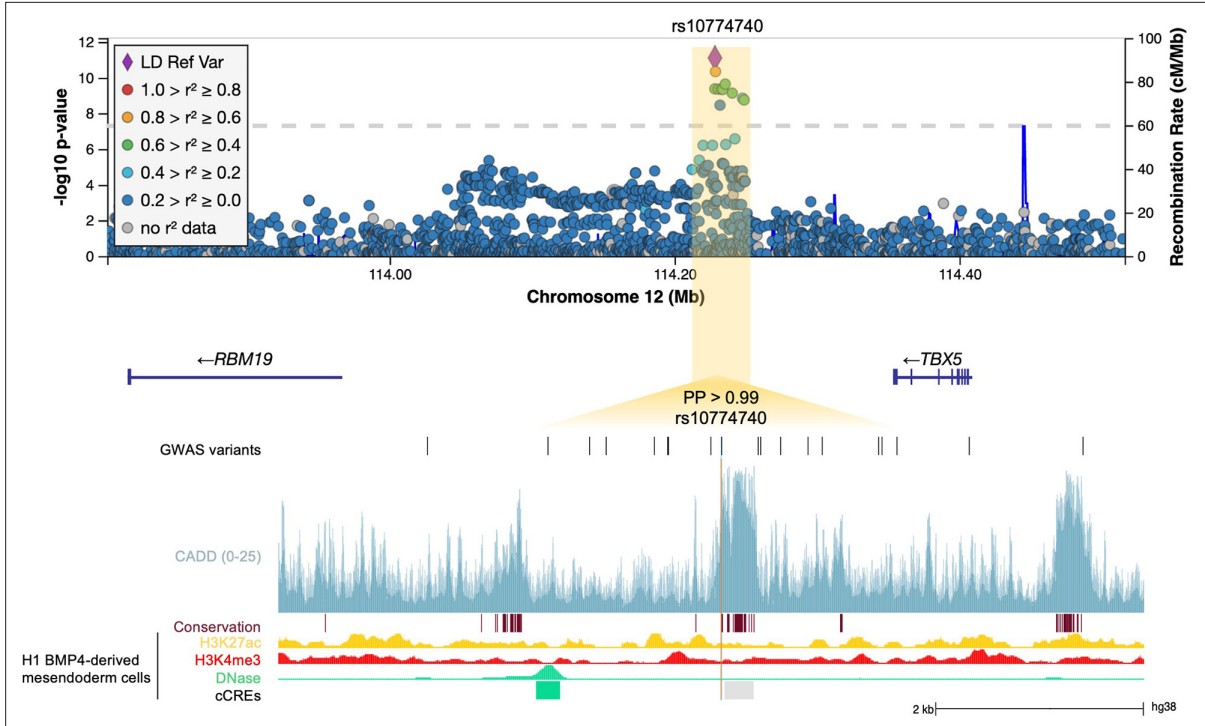

**Figure 3.** 12q24.21. Regional association plot with chromosomal position (GRCh38) denoted along the x axis and strength of association using a –log$_{10}$(p) scale on the y axis. The lead variant (rs10774740) is represented by a purple diamond. Variants are coloured based on their linkage disequilibrium (LD) with the lead variant using 1000 Genomes data from all population groups. Functional annotation of the lead prioritized variant rs10774740 is shown, intersecting with CADD score (version 1.6), PhastCons conserved elements from 100 vertebrates, and ENCODE H3K27ac ChIP-seq, H3K4me3 ChIP-seq, and DNase-seq from mesendoderm cells. ENCODE cCREs active in mesendoderm are represented by shaded boxes; low DNase (grey), DNase-only (green). Genome-wide association study (GWAS) variants with p<0.05 are shown. Note that rs10774740 has a relatively high CADD score for a non-coding variant and intersects with a highly conserved region. PP, posterior probability derived using PAINTOR; cCRE, candidate *cis*-regulatory element.

The online version of this article includes the following figure supplement(s) for figure 3:

**Figure supplement 1.** Heatmap of Hi-C interactions from H1 BMP4-derived mesendoderm cells demonstrating that rs10774740 is located within the same topologically associating domain (TAD) as *TBX5*.

**Figure supplement 2.** Circos plot illustrating significant chromatin interactions between 12q24.21 and the promoter of *TBX5*.

ancestry-specific allele frequencies, effect sizes, and directions for the two lead variants. rs10774740 (T) had a higher allele frequency in individuals of African ancestry (MAF 0.74) compared with European (MAF 0.37) and South Asian (MAF 0.35) populations (*Figure 6—figure supplement 3*), however the effect size and direction were similar between the groups (*Figure 6—figure supplement 4*). rs144171242 (G) was present at a lower allele frequency in South Asian (MAF 0.002) compared with European (MAF 0.008) individuals and was not seen in the African ancestry group (*Figure 6—figure supplement 3*). The effect size of this rare variant was higher in the South Asian than European population (*Figure 6—figure supplement 4*), which may explain why it only reached genome-wide significance after inclusion of South Asian individuals. Finally, comparison with population allele frequencies from gnomAD (*Karczewski et al., 2020*) demonstrated that although there is large variation in the allele frequency of rs10774740 between ancestries this is away from, not towards, the case allele frequency and suggests that the detected associations are not being driven by differences in allele frequency between populations (*Figure 6—figure supplement 3*).

## Fine mapping predicts lead variants to be likely causal

WGS enables further interrogation of loci of interest at high resolution. We therefore repeated the diverse ancestry analysis at each genome-wide significant locus using all variants with minor allele count (MAC)≥3, to determine whether additional ultra-rare variants might be driving the observed association signals. Both rs10774740 at 12q24.21 and rs144171242 at 6p21.1 remained most strongly

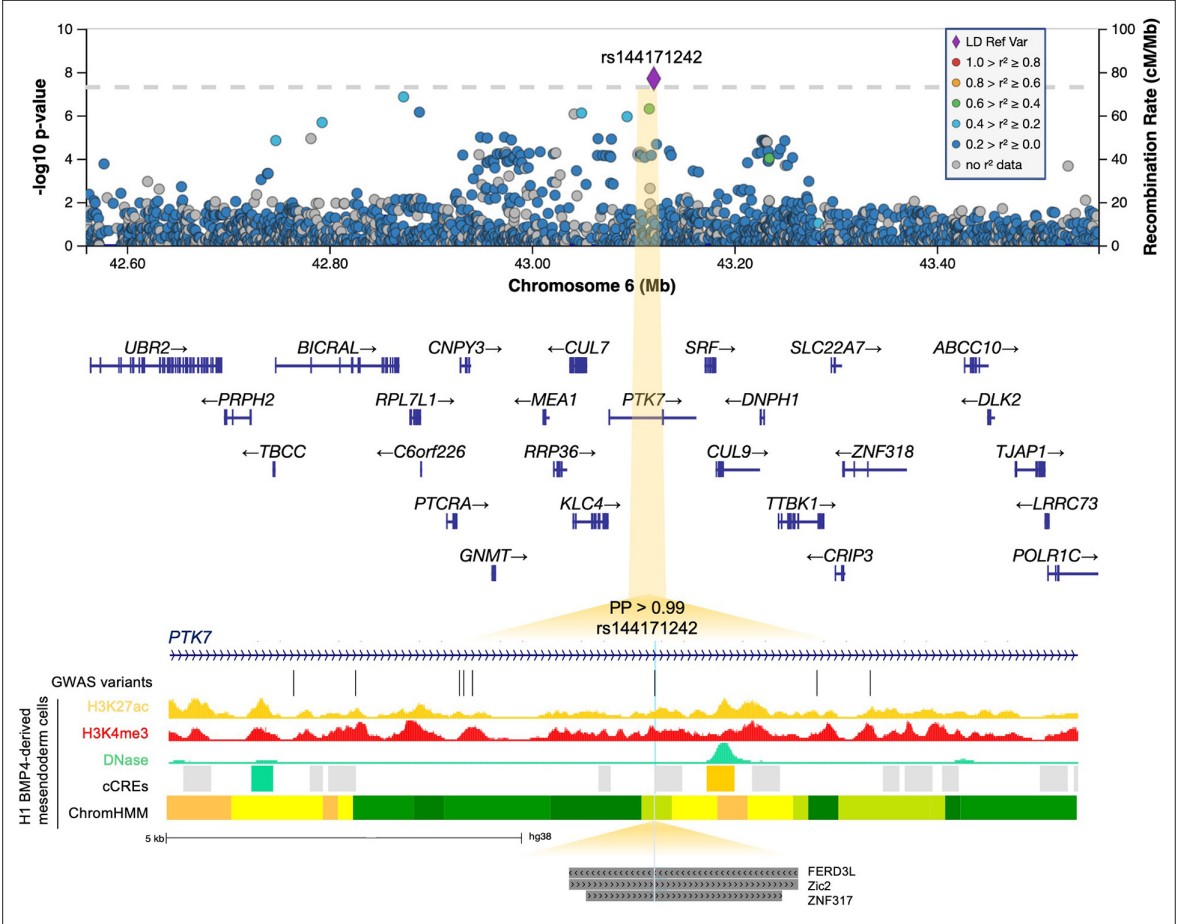

**Figure 4.** 6p21.1. Regional association plot with chromosomal position (GRCh38) along the x axis and strength of association using a –log$_{10}$(p) scale on the y axis. The lead variant (rs144171242) is represented by a purple diamond. Variants are coloured based on their linkage disequilibrium (LD) with the lead variant using 1000 Genomes data from all population groups. Functional annotation of the lead prioritized variant rs144171242 is shown intersecting with ENCODE H3K27ac ChIP-seq, H3K4me3 ChIP-seq, and DNase-seq from mesendoderm cells. ENCODE cCREs active in mesendoderm are represented by shaded boxes; low DNase (grey), DNase-only (green), and distal enhancer-like (orange). ChromHMM illustrates predicted chromatin states using Roadmap Epigenomics imputed 25-state model for mesendoderm cells; active enhancer (orange), weak enhancer (yellow), strong transcription (green), transcribed and weak enhancer (lime green). Predicted transcription factor-binding sites (TFBS) from the JASPAR 2020 CORE collection (**Fornes et al., 2020**) are indicated by dark grey shaded boxes. Genome-wide association study (GWAS) variants with p<0.05 are shown. Note that rs144171242 intersects with both a predicted regulatory region and TFBS. PP, posterior probability derived using PAINTOR; cCREs, candidate *cis*-regulatory elements.

The online version of this article includes the following figure supplement(s) for figure 4:

**Figure supplement 1.** Sequence logos representing the DNA-binding motifs of transcription factors FERD3L and ZNF317.

associated, suggesting they are likely to be causal. Comparison of the different LD patterns seen across African, European, and South Asian population groups at these loci demonstrate how a combined ancestry approach can leverage differences in LD to improve the fine mapping of causal variants (*Figure 6—figure supplement 5*).

We next applied the Bayesian fine-mapping tool PAINTOR (*Kichaev et al., 2020*) which integrates the strength of association, LD patterns, and functional annotations to derive the posterior probability (PP) of a variant being causal. Using this alternative statistical approach, both lead variants were identified as having high probability of being causal: rs10774740 (PP with no annotations 0.77, PP with annotations >0.99) and rs144171242 (PP with no annotations 0.83, PP with annotations >0.99). Conservation and ChIP-seq transcription factor (TF)-binding clusters had the largest impact on PP at 12q24.21 and 6p21.1, respectively. Validation of the lead variants using statistical fine mapping illustrates how the increased sensitivity and improved resolution of WGS compared with genotyping arrays may permit the direct identification of underlying causal variants, particularly in the context of

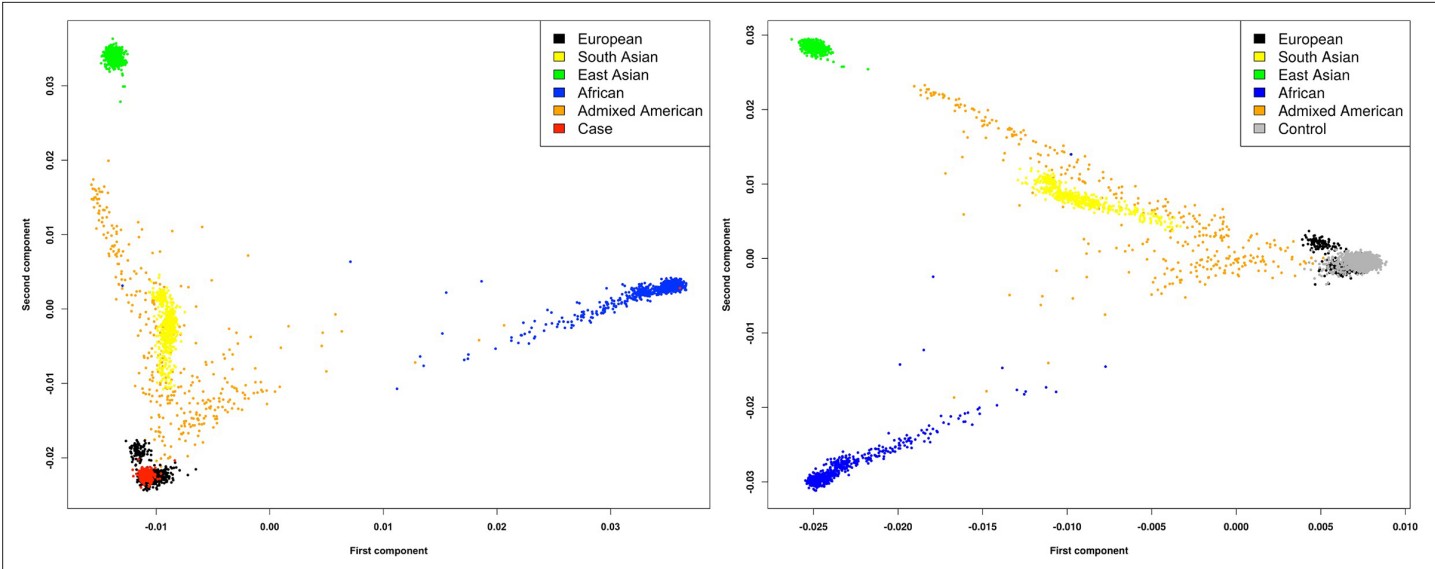

**Figure 5.** Principal component analysis for the replication cohort. Principal component analysis showing the first two principal components for a subset of cases (red) for whom genome-wide genotyping data was available (n=204), and the control (grey) cohort from 100,000 Genomes Project (100KGP) (n=4151) projected onto samples from the 1000 Genomes Project (Phase 3). Both cases and controls had confirmed European ancestry.

examining rarer variants and non-European populations for which imputation performance may be limiting (*Höglund et al., 2019*; *Peterson et al., 2019*).

## Functional annotation implicates *TBX5* and *PTK7*

To explore the functional relevance of these loci, we next interrogated publicly available datasets via UCSC Genome Browser (*Kent et al., 2002*) and used Functional Mapping and Annotation (FUMA; *Watanabe et al., 2017*) to prioritize candidate genes. Given the urinary tract is derived from both embryonic mesoderm and endoderm, where possible we used experimental data obtained from male H1 BMP4-derived mesendoderm cultured cells.

The common, non-coding, intergenic lead variant (rs10774740) at the 12q24.21 locus is predicted to be deleterious (combined annotation-dependent depletion CADD [*Rentzsch et al., 2019*] score 15.54) and intersects with a conserved element (chr12:114228397–114228414; logarithm of odds score 33) that is suggestive of a putative TF-binding site (TFBS; *Figure 3*), however review of experimentally defined TF-binding profiles (*Fornes et al., 2020*) did not identify any known interactions with DNA-binding motifs at this position. Interrogation of epigenomic data from ENCODE (*Moore et al., 2020*) revealed rs10774740 is located ~35 bp away from a cCRE (EH38E1646218), which although has low DNase activity in mesendoderm cells, displays a distal enhancer-like signature (dELS) in cardiac myocytes. There are no known *cis*-eQTL associations with rs10774740, but using experimental Hi-C data generated from H1 BMP4-derived mesendoderm cells (*Dixon et al., 2015*; *Schmitt et al., 2016*) we were able to determine that this locus is within the same topologically associated domain (TAD) as the TF *TBX5* (*Figure 3—figure supplement 1*). Chromatin interaction data mapped this intergenic locus to the promoter of *TBX5* (false discovery rate [FDR] $2.80 \times 10^{-13}$, *Figure 3—figure supplement 2*).

At the 6p21.1 locus, the non-coding lead variant (rs144171242) is in an intron of the inactive tyrosine kinase *PTK7*. This rare variant has a low CADD score (0.93) and lacks any relevant chromatin interaction or known eQTL associations, which is not unexpected given its rarity precludes detection by expression-array experiments. Interrogation of epigenomic annotations from ENCODE (*Moore et al., 2020*) revealed that rs144171242 intersects a cCRE (EH38E2468259) with low DNase activity in mesendoderm cells, but with a dELS in neurons (*Figure 4*). NIH Roadmap Epigenomics Consortium (*Kundaje et al., 2015*) data suggests that rs144171242 may have regulatory activity in mesendoderm cells, classifying this region as transcribed/weak enhancer (12TxEnhW) using the imputed ChromHMM 25-chromatin state model (*Figure 4*). In addition, interrogation of the JASPAR

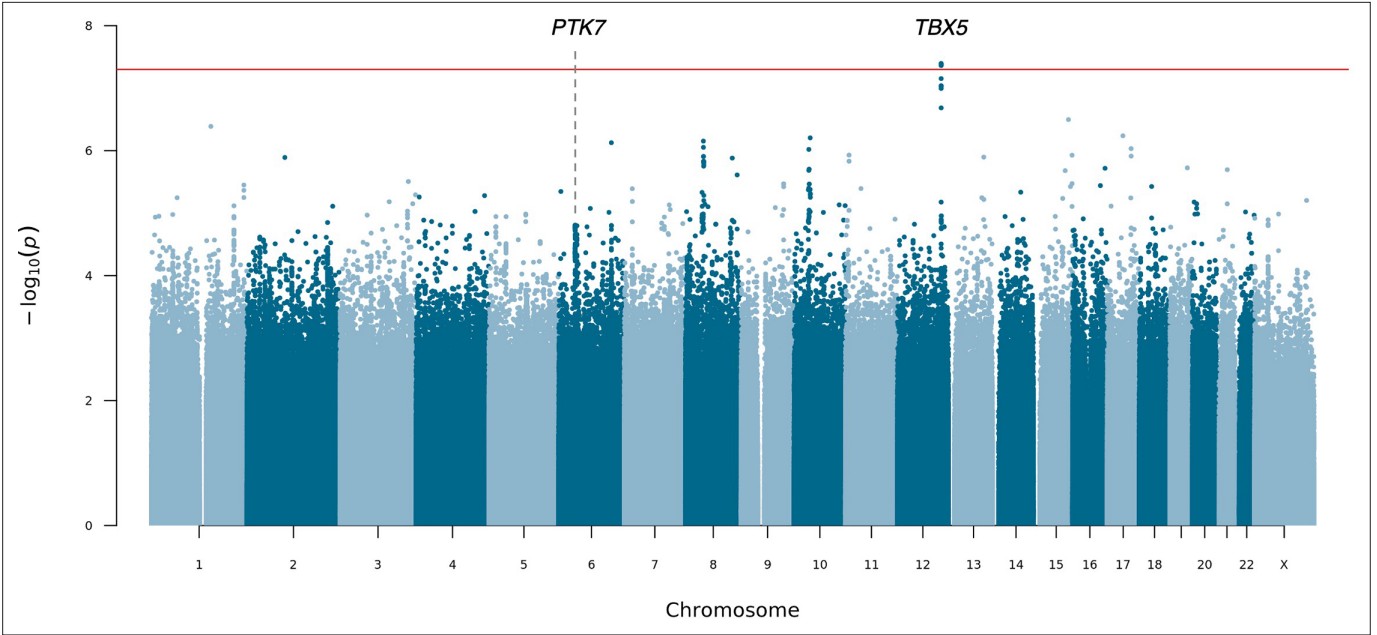

**Figure 6.** Manhattan plot for European sequencing-based genome-wide association study (seqGWAS). A genome-wide single-variant association study was carried out in 88 cases and 17,993 controls for 16,938,500 variants with MAF ≥0.1%. All cases and controls had genetically determined European ancestry. Chromosomal position (GRCh38) is denoted along the x axis and strength of association using a $-\log_{10}$(p) scale on the y axis. Each dot represents a variant. The red line indicates the Bonferroni adjusted threshold for genome-wide significance (p<5 × 10$^{-8}$). The gene in closest proximity to the lead variant at significant loci are listed.

The online version of this article includes the following source data and figure supplement(s) for figure 6:

**Source data 1.** Comparison of diverse ancestry and European genome-wide association study (GWAS) association statistics.

**Figure supplement 1.** Quantile-quantile (Q-Q) plot displaying the observed vs. the expected $-\log_{10}$(p) for each variant tested.

**Figure supplement 2.** Comparison of (**A**) $-\log_{10}$(p) and (**B**) BETA from the diverse ancestry and European-only genome-wide association study (GWAS).

**Figure supplement 3.** Ancestry-specific minor allele frequencies for (**A**) rs10774740 (**T**) at 12q24.21 and (**B**) rs144171242 (**G**) at 6p21.1.

**Figure supplement 4.** Forest plots demonstrating ancestry-specific odds ratios for (**A**) rs10774740 (**T**) and (**B**) rs144171242 (**G**).

**Figure supplement 5.** Linkage disequilibrium (LD) plots for 503 European (EUR), 489 South Asian (SAS), and 661 African (AFR) ancestry individuals from the 1000 Genomes Project (Phase 3).

2020 (*Fornes et al., 2020*) database of experimentally defined TF-binding profiles revealed that rs144171242 intersects with the DNA-binding motifs of FERD3L, ZNF317, and Zic2 (*Figure 4*), suggesting that rs144171242 may potentially affect *PTK7* expression via disruption of TF binding (*Figure 4—figure supplement 1*).

## rs10774740 is associated with prostate cancer and female genitourinary phenotypes

Interrogation of the NHGR/EBI GWAS Catalog (*MacArthur et al., 2017*) revealed that the risk allele rs10774740 (G) is associated with prostate cancer aggressiveness (*Berndt et al., 2015*; p=3 × 10$^{-10}$; OR 1.14; 95% CI 1.09–1.18). PheWAS data from the UK Biobank demonstrated that the protective allele rs10774740 (T) also has a protective effect in female genitourinary phenotypes (*Figure 7*): urinary incontinence (p=8.3 × 10$^{-12}$; OR 0.90; 95% CI 0.87–0.92), female stress incontinence (p=7.9 × 10$^{-10}$; OR 0.89; 95% CI 0.85–0.92), genital prolapse (p=1.1 × 10$^{-9}$; OR 0.92; CI 0.89–0.94) and symptoms involving the female genital tract (p=1.7 × 10$^{-8}$; OR 0.90; 95% CI 0.87–0.94). No significant phenotypic associations affecting other organ systems were seen. These data therefore provide independent validation of a role for *TBX5* in urogenital phenotypes. No known GWAS or PheWAS associations were identified for rs144171242, which is likely too rare to be genotyped or imputed in previous GWAS.

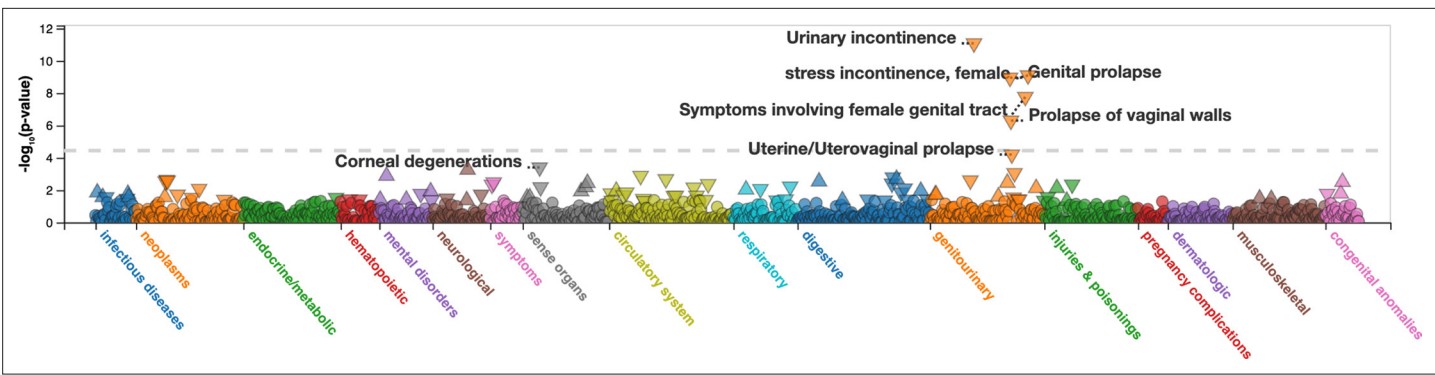

**Figure 7.** Manhattan plot of PheWAS for rs10774740 (**T**) at 12q24.21. Plot downloaded from https://pheweb.org/UKB-SAIGE. The PheWAS was performed using imputed data from ~400,000 White British participants in the UK Biobank using SAIGE. The triangles indicate the direction of effect. The dashed grey line indicates a Bonferroni adjusted significance level of p< 3.6 x 10$^{-5}$ (1403 phenotype codes).

## TBX5 and PTK7 proteins are detected in the developing urinary tract

To determine whether TBX5 and PTK7 proteins can be detected during urinary tract development, immunohistochemistry was undertaken in a 7-week gestation normal human embryo (*Figure 8*). At this stage of development, the urogenital sinus is a tube composed of epithelia that will differentiate into urothelial cells of the proximal urethra and the urinary bladder. Uroplakin 1B, a water-proofing protein, was detected in urogenital sinus epithelia (*Figure 8B*). PTK7 was detected in epithelia lining the urogenital sinus, and intensely in stromal-like cells surrounding the mesonephric ducts (*Figure 8C*). TBX5 was detected in a nuclear pattern in a subset of epithelial cells lining the urogenital sinus (*Figure 8D*). Omission of primary antibodies resulted in absent signals, as expected (*Figure 8E*). Immunostaining of a second human embryo at the same stage of gestation produced similar results, with a prominent signal for PTK7 in stromal cells surrounding the mesonephric duct and a subset of nuclei in the epithelium of the urogenital sinus stained for TBX5 (*Figure 8—figure supplement 1*).

## Monogenic causes of PUV are uncommon

Having identified two novel gene associations through single-variant association testing, we next aimed to determine whether there was any gene-based enrichment of rare coding variation in this PUV cohort. Single-variant association tests can be underpowered when variants are rare and collapsing variant data into specific regions or genes can increase power and aid gene discovery.

We therefore aggregated rare (gnomAD allele frequency [AF]<0.1% [*Karczewski et al., 2020*]), predicted deleterious [protein-truncating, or CADD ≥20 [*Rentzsch et al., 2019*]] SNVs and small indels by gene, comparing the burden between cases and controls on an exome-wide basis. No significant enrichment was detected in any of the 19,161 protein-coding genes analysed after correction for multiple testing (*Figure 9* and *Figure 9—figure supplement 1*). The median number of variants tested per gene was 41 (IQR 47). None of the genes previously associated with congenital bladder outflow obstruction showed evidence of enrichment: *BNC2* (congenital lower urinary tract obstruction, MIM 618612); *FLNA* (frontometaphyseal dysplasia, MIM 305620); *HPSE2* and *LRIG2* (urofacial syndrome, MIM 236730/615112); *CHRM3* (prune-belly syndrome, MIM 100100); *MYOCD* (megabladder, MIM 618719). Despite the limited power of this cohort, the lack of gene-based enrichment seen is consistent with an absence of proven Mendelian causes of non-syndromic PUV.

## Structural variation affecting *cis*-regulatory elements is enriched

Large, rare CNVs have been identified in patients with PUV using conventional microarrays (*Boghossian et al., 2016*; *Caruana et al., 2016*; *Faure et al., 2016*; *Verbitsky et al., 2019*), however high-coverage WGS enables detection of smaller structural variants (SVs) with superior resolution (*Gross et al., 2019*; *Zhou et al., 2016c*), and allows the identification of balanced rearrangements including inversions. We therefore aimed to test for association with different types of SVs, by comparing the burden of rare (MAF <0.1%), autosomal SVs on an exome-wide and *cis*-regulatory element basis.

We first focused our analysis on rare SVs that were potentially gene-disrupting by extracting those that intersected with at least one exon. Although we observed an increased burden of all

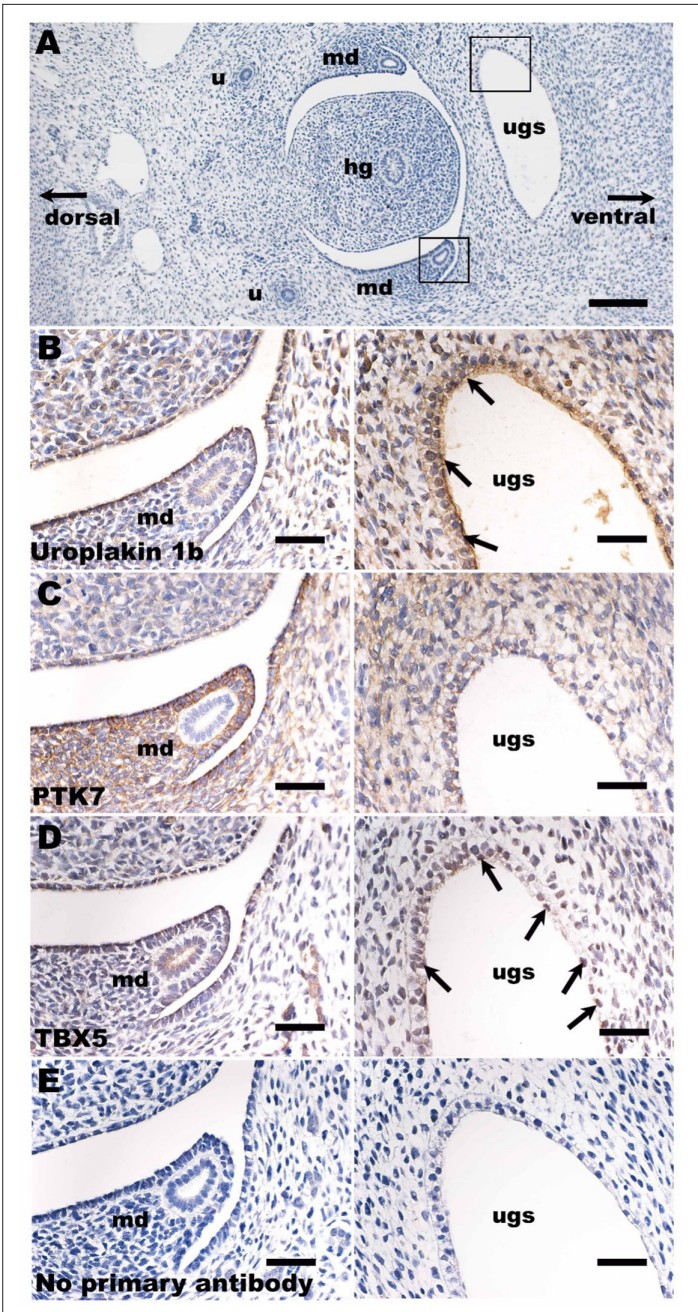

**Figure 8.** Immunohistochemistry in human embryogenesis. (**A**) Overview of transverse section of a normal human embryo 7 weeks after fertilization. The section has been stained with haematoxylin (blue nuclei). Boxes around the urogenital sinus and the mesonephric duct mark similar areas depicted under high power in (**B–E**). In (**B–D**), sections were reacted with primary antibodies, as indicated; in (**E**) the primary antibody was omitted. (**B–E**) were counterstained with haematoxylin. In (**B–E**), the left-hand frame shows the region around the mesonephric duct, while the right-hand frame shows one lateral horn of the urogenital sinus. (**B**) Uroplakin 1b immunostaining revealed positive signal (brown) in the apical aspect of epithelia lining the urogenital sinus (arrows, right frame), the precursor of the urinary bladder and proximal urethra. Uroplakin 1b was also detected in the flat monolayer of mesothelial cells (left frame) that line the body cavity above the mesonephric duct. (**C**) There were strong PTK7 signals (brown cytoplasmic staining) in stromal-like cells around the mesonephric duct (left frame), whereas the epithelia of the duct itself were negative. PTK7 was also detected in a reticular pattern in epithelia lining the urogenital sinus (right frame) and in stromal cells near the sinus. (**D**) A subset of epithelial cells lining the urogenital sinus (right frame) immunostained for TBX5 (brown nuclei; some are arrowed). The mesothelial cells near the mesonephric duct (left frame) were also positive for TBX5. (**E**) This negative control section had the primary

*Figure 8 continued on next page*

*Figure 8 continued*

antibody omitted; no specific (brown) signal was noted. Bar is 400 µm in (**A**), and bars are 100 µm in (**B–E**). ugs, urogenital sinus; md, mesonephric duct; hg, hindgut; u, ureter.

The online version of this article includes the following figure supplement(s) for figure 8:

**Figure supplement 1.** Immunohistochemistry of a second 7-week human embryo counterstained with haematoxylin.

SV types in cases compared with controls, this only reached statistical significance for inversions (p=2.1 × 10$^{-3}$) when corrected for the multiple SV comparisons performed (*Table 3*). No difference in SV size between the cohorts was seen. Furthermore, exome-wide gene-based burden analysis did not detect any gene-level enrichment of rare SVs overall or when stratified by type, indicating that, in this cohort at least, rare structural variation does not appear to affect any single gene recurrently.

Given the tightly controlled transcriptional networks that govern embryogenesis, we hypothesized that regulatory regions may be preferentially affected by rare structural variation. To investigate this, we identified rare (MAF <0.1%), autosomal SVs that intersected with 926,535 genome-wide cCREs curated by ENCODE (*Moore et al., 2020*). A significant enrichment of cCRE-intersecting SVs was observed for inversions (61.4% vs. 47.1%, p=1.2 × 10$^{-3}$) and duplications (78.8% vs. 67.5%, p=5.0 × 10$^{-3}$) (*Table 3*). While the median size of inversions was larger in cases, this was not statistically significant (129 kb vs. 94 kb, p=0.12).

To further characterize this enrichment, we repeated the burden analysis stratifying by cCRE subtype (distal enhancer-like signature [dELS], proximal enhancer-like signature [pELS], promoter-like signature [PLS], CTCF-only and DNase-H3K4me3) and demonstrated a consistent signal across all cCREs for inversions (*Figure 10*), most significantly affecting CTCF-only elements (49.2% vs. 31.7%, p=3.1 × 10$^{-5}$, *Figure 10—source data 1*). These elements act as chromatin loop anchors suggesting that inversions affecting these regions may potentially alter long-range regulatory mechanisms mediated by chromatin conformation. Duplications affecting pELS elements were also significantly enriched in cases (29.5% vs. 16.8%, p=2.7 × 10$^{-4}$).

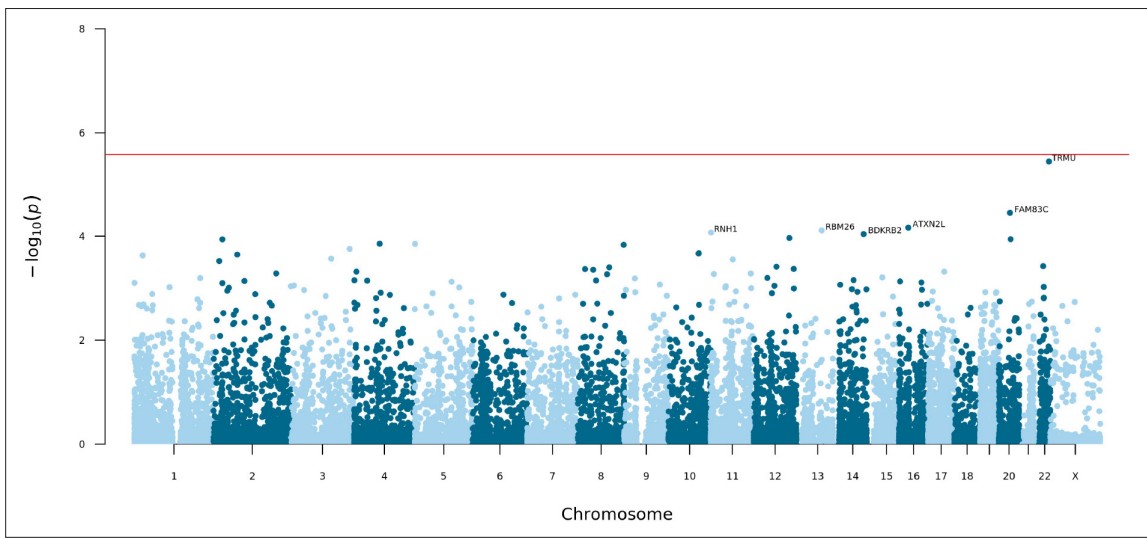

**Figure 9.** Manhattan plot of exome-wide rare coding variant analysis. Chromosomal position (GRCh38) is shown on the x axis and strength of association using a –log$_{10}$(p) scale on the y axis. Each dot represents a gene. The red line indicates the Bonferroni adjusted threshold for exome-wide significance (p=2.58 × 10$^{-6}$). Genes with p<10$^{-4}$ are labelled.

The online version of this article includes the following source data and figure supplement(s) for figure 9:

**Source data 1.** Exome-wide rare coding variant analysis by gene.

**Figure supplement 1.** Quantile-quantile (Q-Q) plot displaying the observed vs. the expected –log$_{10}$(p) for each gene tested.

**Table 3. Structural variant burden analysis.**
The burden of rare autosomal structural variants intersecting with (a) at least one exon or (b) a *cis*-regulatory element was compared between 132 cases and 23,727 controls. cCRE, candidate *cis*-regulatory element; PUV, posterior urethral valves; CNV, copy number variant; DEL, deletion; DUP, duplication; INV, inversion; OR, odds ratio; CI, 95% confidence interval, IQR, interquartile range.

| | | EXON | | cCRE | |
|---|---|---|---|---|---|
| | | PUV | Controls | PUV | Controls |
| CNV | n (%) | 109 (82.6) | 17,961 (75.7) | 111 (84.1) | 18,773 (79.1) |
| | OR (CI) | 1.52 (0.96–2.50) | | 1.39 (0.87–2.34) | |
| | Fisher's exact p | 0.07 | | 0.2 | |
| | Median size (kb) (IQR) | 104 (183) | 94 (158) | 80 (165) | 80 (128) |
| | p (Wilcoxon) | 0.75 | | 0.75 | |
| DEL | n (%) | 117 (88.6) | 19,987 (84.2) | 132 (100) | 23,031 (97.1) |
| | OR (CI) | 1.46 (0.85–2.69) | | Inf | |
| | Fisher's exact p | 0.19 | | 0.04 | |
| | Median size (kb) (IQR) | 1.4 (4.2) | 1.8 (4.6) | 1.5 (4.1) | 1.9 (4.5) |
| | p (Wilcoxon) | 0.18 | | $4.1 \times 10^{-4}$ | |
| DUP | n (%) | 59 (44.7) | 8,476 (35.7) | 104 (78.8) | 16,004 (67.5) |
| | OR (CI) | 1.45 (1.01–2.08) | | 1.79 (1.17–2.83) | |
| | Fisher's exact p | 0.04 | | $5.0 \times 10^{-3}$ | |
| | Median size (kb) (IQR) | 3.7 (5.7) | 3.0 (5.6) | 2.0 (5.3) | 2.3 (5.3) |
| | p (Wilcoxon) | 0.16 | | 0.49 | |
| INV | n (%) | 66 (50.0) | 8,736 (36.8) | 81 (61.4) | 11,171 (47.1) |
| | OR (CI) | 1.72 (1.20–2.45) | | 1.79 (1.24–2.59) | |
| | Fisher's exact p | $2.1 \times 10^{-3}$ | | $1.2 \times 10^{-3}$ | |
| | Median size (kb) (IQR) | 253 (1931) | 261 (1642) | 129 (459) | 94 (779) |
| | p (Wilcoxon) | 0.44 | | 0.12 | |

## Discussion

Using a seqGWAS we have identified the first robust genetic associations of PUV, implicating *TBX5* and *PTK7* in its underlying pathogenesis. We detected an enrichment of rare structural variation affecting cCREs and demonstrate that monogenic causes of PUV are not a common feature, at least within this cohort. We also show that a well-controlled diverse ancestry WGS approach can increase power for disease locus discovery in rare disease and facilitate the fine mapping of causal variants.

The majority of genetic association studies are carried out in individuals of European ancestry, however with WGS allowing unbiased variant detection combined with the use of appropriate statistical methodology to mitigate confounding by population structure, it is widely recognized that increasing ancestral diversity in genetic studies is scientifically and ethically necessary (*Fatumo et al., 2022*; *Peterson et al., 2019*). On this basis we opted to combine individuals regardless of ancestral background, including a greater proportion of non-White individuals than is present in the UK (87% individuals are recorded as 'White' in the 2011 UK census [*General Register Office for Scotland, Northern Ireland Statistics and Research Agency, Office for National Statistics, Census Division, 2020*]), suggesting mitigation of the Euro-centric bias present in most previous genomic studies, at least with respect to the UK population. We used a generalized mixed model association test with saddlepoint approximation to maximize the signal from the resulting diverse ancestry, case-control imbalanced dataset.

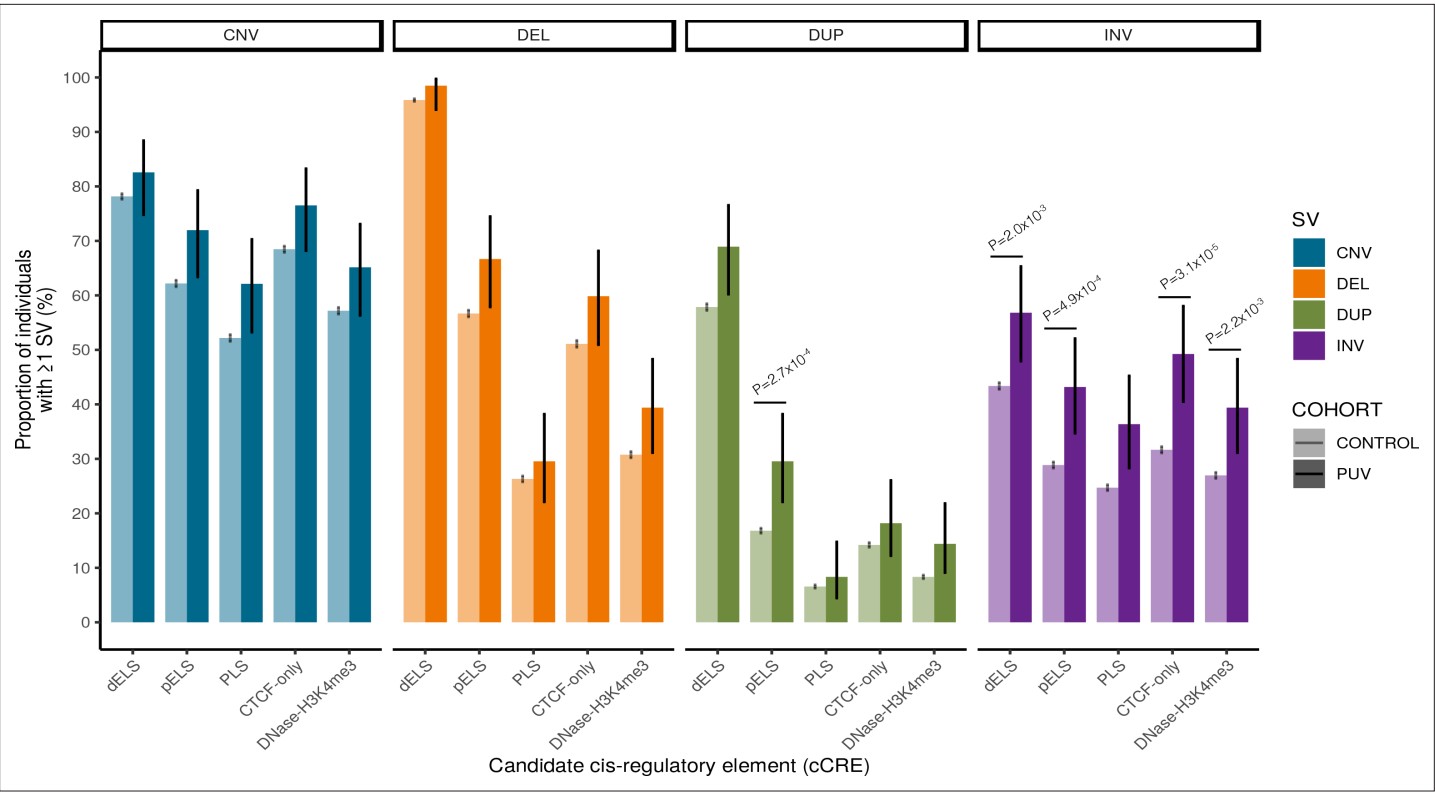

**Figure 10.** Rare structural variant burden analysis. The proportion of individuals with ≥1 rare autosomal structural variant intersecting with an ENCODE candidate *cis*-regulatory element (cCRE) in cases and controls was enumerated using a two-sided Fisher's exact test. Note that inversions affecting cCRE are enriched in PUV. Vertical black bars indicate 95% confidence intervals. Unadjusted p-values shown are significant after correction for multiple testing (p<2.5 × 10⁻³). CNV, copy number variant; DEL, deletion; DUP, duplication; INV, inversion; PUV, posterior urethral valves; dELS, distal enhancer-like signature; pELS, proximal enhancer-like signature; PLS, promoter-like signature; cCRE, candidate *cis*-regulatory element.

The online version of this article includes the following source data for figure 10:

**Source data 1.** Structural variant cCRE analysis.

We identified a significant protective effect of the common variant rs10774740 (T), highlighting that common variants can contribute to an individual's risk of a rare disease, as is increasingly being recognized (*Niemi et al., 2018*). The effect size and direction were consistent between African, European, and South Asian ancestries, despite differences in allele frequency between the population groups. Using experimentally determined chromatin interaction data from mesendoderm cells, we mapped this locus to the promoter of the transcription factor *TBX5*, rare variants of which cause autosomal-dominant Holt-Oram syndrome (MIM 142900), characterized by congenital cardiac septal defects and upper-limb anomalies (*Basson et al., 1997*; *Li et al., 1997*).

The association of the risk allele (G) with prostate cancer aggressiveness in men and genital prolapse and urinary incontinence in women raises the intriguing possibility that *TBX5*, which shows moderate expression in the adult bladder, is also associated with lower urinary tract phenotypes in adults. These findings are consistent with the observation that variation in genes associated with other developmental anomalies have been linked to malignancy in the same organ (*Su et al., 2012*; *Sun et al., 2017*), highlighting the shared molecular pathways driving both embryogenesis and cancer.

We also identified an association of the rare variant rs144171242 with PUV, located in an intron of *PTK7* and predicted to have regulatory activity in mesendoderm cells. Given its rarity, this association would not have been detected using a conventional array-based GWAS approach, highlighting the utility of seqGWAS. The inclusion of South Asian individuals, in whom the effect size of rs144171242 is larger, increased our power to detect association which was not genome-wide significant in the European-only analysis. PTK7 (protein tyrosine kinase 7) is an evolutionarily conserved transmembrane receptor required for vertebrate embryonic patterning and morphogenesis, and a key regulator of planar cell polarity (PCP) via the non-canonical Wnt pathway (*Berger et al., 2017*). The PCP pathway

is critical for determining the orientation of cells in the plane of an epithelium, regulating a process called convergent extension whereby cells intercalate by converging in one axis and elongating in the perpendicular axis. Altered expression of *PTK7* was initially observed in cancer (**Dunn and Tolwinski, 2016**; **Mossie et al., 1995**), but rare missense variants in *PTK7* have since been linked to neural tube defects (**Lei et al., 2019**; **Wang et al., 2015**) and scoliosis (**Hayes et al., 2014**) in both humans and animal models, confirming a role in embryonic development.

Aberrations of mesonephric duct and urogenital sinus maturation have been implicated in the pathogenesis of PUV (**Krishnan et al., 2006**). Our observations that TBX5 and PTK7 are present at the appropriate timepoint during normal human embryogenesis, in or around the mesonephric ducts and urogenital sinus, suggest that altered regulation of expression of either may perturb the normal development of these structures. Further functional studies will however be necessary to provide definitive evidence of a role for these proteins in urinary tract development. Interestingly, mesoderm-specific conditional deletion of *Ptk7* in mice has been shown to affect convergent extension and tubular morphogenesis of the mesonephric duct at E18.5, leading to a shortened duct with increased diameter and reduced coiling, resulting in male sterility (**Xu et al., 2018**; **Xu et al., 2016**). Whether similar disruption in mesonephric duct morphogenesis is seen at E14 (corresponding to development of the urethra) remains to be seen but may provide further insights into the biological mechanisms underpinning PUV.

PUV is a male-limited phenotype, although urethral abnormalities have been reported in the female relatives of affected males (**Kolvenbach et al., 2019**). This sexual dimorphism is most likely the result of anatomical differences in the development and length of the urethra between males and females, with the association between rs10774740 at the *TBX5* locus and female genital prolapse and urinary incontinence suggesting that females may also manifest lower urinary tract phenotypes. While X-linked inheritance has been reported in two patients with syndromic PUV (**Wade et al., 2021**), we detected no significant common or rare variant associations on chromosome X in this cohort to suggest that genetics underpin the male specific nature of this condition.

Rare CNVs, detected using a microarray-based approach, have recently been shown to be enriched in patients with kidney and urinary tract anomalies, but were not found to be enriched in 141 PUV patients (**Verbitsky et al., 2019**). Similarly, we did not identify an increased burden of rare CNVs in individuals with PUV, but we did observe a higher number of rare, exonic CNVs than Verbitsky et al. (2019) (82.6% vs. 32.6%), most likely reflecting the increased sensitivity and resolution of WGS for SV detection as well as the difference in size threshold for inclusion (>10 kb compared with >100 kb). Importantly, none of the CNVs recurrently affected a particular gene in our cohort which, in combination with the lack of gene-based enrichment seen in our rare SNV/indel analysis, suggests that monogenic causes of PUV are rare, although our sample size would be underpowered to detect significant genetic heterogeneity.

Intriguingly, we demonstrated an enrichment of rare duplications and inversions affecting cCREs. Genomic duplications have previously been linked with PUV (**Schierbaum et al., 2021**; **Verbitsky et al., 2019**) but this is the first study to examine balanced inversions in a PUV cohort. Current understanding of the functional relevance of inversions is limited as the balanced nature and location of breakpoints within complex repeat regions make detection challenging (**Puig et al., 2015**). Inversions can directly disrupt coding sequences or regulatory elements, as well as predispose to other SVs, and have been associated with hemophilia A (**Lakich et al., 1993**), Hunter syndrome (**Bondeson et al., 2016**), neurodegenerative (**Webb et al., 2008**) and autoimmune disease (**Namjou et al., 2021**; **Salm et al., 2012**). The strongest signal we observed was for inversions affecting CTCF-only regions, potentially implicating disrupted chromatin looping in the underlying pathogenesis of PUV and raising the interesting possibility that non-specific perturbation of long-range regulatory networks or TADs could manifest as PUV, perhaps due to sensitivity of integration of the mesonephric duct into the posterior urethra to even minor abnormalities of gene expression. While the SV analyses described here generate some interesting hypotheses, these findings should be considered exploratory until they can be replicated in an independent cohort.

In summary, we have shown that both rare and common variation can contribute to an individual's risk of PUV, and that structural variation affecting regulatory networks may also play a role. Given that PUV does not usually exhibit a Mendelian pattern of inheritance, our data provides support for a complex genetic architecture for this rare disorder, where different types of genetic variation across

the allele frequency spectrum may interact to influence susceptibility. Better powered studies will be needed to identify additional genetic risk factors with smaller effect sizes and to explore how these variants might be influenced by maternal or in utero factors (*Nicolaou et al., 2015*).

This study has several strengths. WGS enables ancestry-independent variant detection, uniform genome-wide coverage, improved SV resolution and detection, as well as direct sequencing of underlying causal variants. Using case-control data from over 20,000 individuals sequenced on the same platform also minimizes confounding by technical artefacts. Inclusion of diverse ancestries increased our power to detect both novel associations and the underlying causal variant, with the lack of genomic inflation and subsequent replication indicating these associations are robust. Furthermore, we integrated seqGWAS, epigenomic and chromatin interaction data to ascertain the functional relevance of loci and identify biologically plausible genes, validated by detection of the relevant proteins at the appropriate site and developmental stage.

A limitation of this study is its relatively low statistical power to detect association due to the limited size of the cohort; sufficient power exists to detect low-frequency variants with large effects (e.g., MAF 1% with OR >7.5), but rare variants, or those with smaller effect sizes, may be missed with this sample size (*Figure 2—figure supplement 2*). Future meta-analyses will be necessary to identify additional loci, as well as expanded efforts to recruit non-Europeans given the low absolute numbers of non-European cases in this study. Furthermore, while WGS offers improved SV resolution over microarrays, false positives may occur and are dependent on the SV calling algorithm used. Ideally, long-read sequencing and independent validation would provide more comprehensive SV detection, especially of larger variants in complex, repetitive and GC-rich regions. Finally, although we have assessed the relevance of the associated loci using bioinformatic approaches and shown that publicly available and our own experimental data support the association, the biological mechanisms linking these genes with PUV have yet to be elucidated.

To our knowledge, this is the first study to utilize diverse ancestry WGS for association testing in a rare disease. Combining WGS data across ancestries increased power to detect the first two robust genetic associations for PUV and identified the likely causal variants through enhanced fine mapping. Finally, integration of functional genomic and experimental data implicated *TBX5* and *PTK7* in the pathogenesis of PUV, providing new insights into the biological pathways underlying this important but poorly understood disorder.

## Methods
### The 100,000 Genomes Project

The Genomics England dataset (version 10) (*Smedley et al., 2021*) consists of WGS data, clinical phenotypes encoded using HPO codes (*Köhler et al., 2021*), and retrospective and prospectively ascertained National Health Service (NHS) hospital records for 89,139 individuals recruited with cancer, rare disease, and their unaffected relatives. Ethical approval for the 100KGP was granted by the Research Ethics Committee for East of England – Cambridge South (REC Ref 14/EE/1112). Written informed consent was obtained from all participants and/or their guardians. *Figure 1* details the study workflow.

Cases were recruited from 13 NHS Genomic Medicine Centers across the UK as part of the 100KGP 'Congenital anomalies of the kidneys and urinary tract (CAKUT)' cohort with the following inclusion criteria: CAKUT with syndromic manifestations in other organ systems; isolated CAKUT with a first-degree relative with CAKUT or unexplained CKD; multiple distinct renal/urinary tract anomalies; CAKUT with unexplained end-stage kidney disease before the age of 25 years. Those with a clinical or molecular diagnosis of ADPKD or ARPKD, or who had a known genetic or chromosomal abnormality were excluded. One-hundred and thirty-six male individuals with a diagnosis of PUV were identified using the HPO term 'HP:0010957 congenital posterior urethral valve'.

All cases underwent assessment via the clinical interpretation arm of the 100KGP to determine a molecular diagnosis. This process involved the examination of protein-truncating and missense variants from an expert-curated panel of 57 CAKUT-associated genes followed by multi-disciplinary review and application of ACMG (*Richards et al., 2015*) criteria to determine pathogenicity. CNVs affecting the 17q12 region (ISCA-37432-Loss), which includes *HNF1B*, were also assessed. No pathogenic/likely pathogenic variants were identified in genes previously associated with congenital bladder outflow

obstruction (*HPSE2, LRIG2, CHRM3, MYOCD, BNC2*). Two pathogenic/likely pathogenic variants affecting the 17q12 locus and *FOXC1* were identified in two individuals, but these were not determined to be causal for PUV (see Appendix 1).

The control cohort consisted of 27,660 unaffected relatives of non-renal rare disease participants, excluding those with HPO terms and/or hospital episode statistics (HES) data consistent with kidney disease or failure. By utilizing a case-control cohort sequenced on the same platform, we aimed to minimize confounding by technical artefacts.

## DNA preparation and extraction

Ninety-nine percent of DNA samples were extracted from blood and prepared using EDTA, with the remaining 1% sourced from saliva, tissue, and fibroblasts. Samples underwent quality control assessment based on concentration, volume, purity, and degradation. Libraries were prepared using the Illumina TruSeq DNA PCR-Free High Throughput Sample Preparation kit or the Illumina TruSeq Nano High Throughput Sample Preparation kit.

## WGS, alignment, and variant calling

Samples were sequenced with 150 bp paired-end reads using an Illumina HiSeq X and processed on the Illumina North Star Version 4 Whole Genome Sequencing Workflow (NSV4, version 2.6.53.23), comprising the iSAAC Aligner (version 03.16.02.19) and Starling Small Variant Caller (version 2.4.7). Samples were aligned to the *Homo sapiens* NCBI GRCh38 assembly. Alignments had to cover ≥95% of the genome at ≥15X with mapping quality >10 for samples to be retained. Samples achieved a mean of 97.4% coverage at 15X with a median genome-wide coverage of 39X. Samples with <2% cross-contamination as determined by the VerifyBamID algorithm were kept. CNV and SV (>50 bp) calling was performed using CANVAS (*Roller et al., 2016*; version 1.3.1) and MANTA (*Chen et al., 2016a*; version 0.28.0), respectively. CANVAS determines coverage and MAF to assign copy number (>10 kb) whereas MANTA combines paired and split-read algorithms to detect SVs (<10 kb).

## gVCF annotation and variant-level quality control

gVCFs were aggregated using gvcfgenotyper (Illumina, version 2019.02.26) with variants normalized and multi-allelic variants decomposed using vt (version 0.57721; *Tan et al., 2015*). Variants were retained if they passed the following filters: missingness ≤5%, median depth ≥10, median GQ ≥15, percentage of heterozygous calls not showing significant allele imbalance for reads supporting the reference and alternate alleles (ABratio) ≥25%, percentage of complete sites (completeGTRatio) ≥50% and p-value for deviations from Hardy-Weinberg equilibrium (HWE) in unrelated samples of inferred European ancestry ≥1 × 10⁻⁵. Male and female subsets were analysed separately for sex chromosome quality control. Per-variant MAC was calculated across the case-control cohort. Annotation was performed using Variant Effect Predictor (version 98.2; *McLaren et al., 2016*) including CADD (version 1.5; *Rentzsch et al., 2019*), and allele frequencies from publicly available databases including gnomAD (version 3; *Karczewski et al., 2020*) and TOPMed (Freeze 5; *Taliun et al., 2021*). Variants were filtered using bcftools (version 1.11; *Danecek et al., 2021*).

## Relatedness estimation and principal component analysis

A set of 127,747 high-quality autosomal LD-pruned bi-allelic SNVs with MAF >1% was generated using PLINK (version 1.9; *Purcell et al., 2007*). SNVs were included if they met all the following criteria: missingness <1%, median GQ ≥30, median depth ≥30, ABratio ≥0.9, completeness ≥0.9. Ambiguous SNVs (A/T or G/C) and those in a region of long-range high LD were excluded. LD pruning was carried out using an r² threshold of 0.1 and window of 500 kb. SNVs out of HWE in any of the AFR, EAS, EUR, or SAS 1000 Genomes populations were removed (pHWE <1 × 10⁻⁵). Using this variant set, a pairwise kinship matrix was generated using the PLINK2 (*Chang et al., 2015*) implementation of the KING-Robust algorithm (*Manichaikul et al., 2010*) and a subset of unrelated samples was ascertained using a kinship coefficient threshold of 0.0884 (second-degree relationships). Two cases and 1354 controls were found to be related by this method and were removed, leaving 134 cases and 26,306 controls. Ten principal components were generated using PLINK2 (*Chang et al., 2015*) for ancestry-matching and for use as covariates in the association analyses.

## Ancestry-matching of cases and controls

Given the mixed-ancestry composition of the cohort, we employed a case-control ancestry-matching algorithm to optimize genomic similarity and minimize the effects of population structure. A custom R script (*Source code 1*) was used to match cases to controls within a distance threshold calculated using the top 10 principal components weighted by the percentage of genetic variation explained by each component (*Figure 1—figure supplement 1*). Only controls within a user-defined specified distance of a case were included with each case having to match a minimum of two controls to be included in the final cohort. A total of two cases and 2579 controls were excluded using this approach, leaving 132 cases and 23,727 controls for further analysis.

For the European-only analysis, a subset of individuals predicted to have ≥0.8 probability of European ancestry inferred by a random forest model (Ntrees = 400) generated using the first eight principal components of the 1000 Genomes Project (Phase 3) data were extracted (89 cases and 19,418 controls). After ancestry-matching 88 cases and 17,993 controls remained for analysis.

## Single-variant seqGWAS

Whole-genome single-variant association analysis was carried out using the R package SAIGE (version 0.42.1) (*Zhou et al., 2016a*) which uses a GLMM to account for population stratification. High-quality, autosomal, bi-allelic, LD-pruned SNVs with MAF >5% were used to generate a genetic relationship matrix and fit the null GLMM. Sex and the top 10 principal components were used as covariates (fixed effects). Sex was used as a covariate to control for the anatomical differences in development of the urethra (and in utero hormonal changes) between the sexes in the control cohort. SNVs and indels with MAF ≥0.1% that passed the following quality control filters were retained: MAC ≥20, missingness <1%, HWE $p>10^{-6}$, and differential missingness $p>10^{-5}$. A score test (*Chen et al., 2016b*) for association was performed for 19,651,224 variants (diverse ancestry single-variant seqGWAS) and 16,938,500 variants (European-only seqGWAS). When case-control ratios are unbalanced, as in our study (1:180), type 1 error rates are inflated because the asymptotic assumptions of logistic regression are invalidated. SAIGE employs a saddlepoint approximation (*Dey et al., 2017*) to calibrate score test statistics and obtain more accurate p-values than the normal distribution.

At each of the genome-wide significant loci, we used SAIGE to perform (a) conditional analysis to identify secondary independent associations and (b) high-resolution single-variant analysis using all variants with MAC ≥3 to ascertain whether the observed signal was being driven by rare variation. Epistasis between the lead variants was assessed using logistic regression in PLINK (version 1.9; *Purcell et al., 2007*). One limitation of SAIGE is that the betas estimated from score tests can be biased at low MACs and therefore odds ratios for variants with MAF <1% were calculated separately using allele counts in R. The R packages qqman (*Turner, 2015*) and GWASTools (*Gogarten et al., 2012*) were used to create Manhattan and Q-Q (quantile-quantile) plots, and LocusZoom (*Pruim et al., 2010*) to visualize regions of interest.

## Replication

The replication cohort consisted of 395 individuals with PUV; 333 recruited from Poland and Germany as part of the CaRE for LUTO (Cause and Risk Evaluation for Lower Urinary Tract Obstruction) Study, and 62 from Manchester, UK. None of the individuals had been recruited to the 100KGP. All were of self-reported European ancestry. To confirm that this cohort was also genetically European, we performed a principal component analysis (using PLINK2) analyzing a subset (n=204) of the Polish and German PUV cases for whom genome-wide genotyping data was available, projecting the first two principal components onto samples from the 1000 Genomes Project (Phase 3) (*Auton et al., 2015*) labelled by population group. KASP (Kompetitive Allele-Specific PCR) genotyping of the lead variants at the top four loci using a threshold of $p<5 \times 10^{-7}$ was carried out: rs10774740 at 12q24.21, rs144171242 at 6p21.1, rs1471950716 at 10q11.21, rs199975325 at 14q21.1. The peri-centromeric location of rs1471950716 at 10q11.21 caused the genotyping assay to fail and another variant with evidence of association (rs137855548; $p=1.46 \times 10^{-6}$) was used instead. The replication control cohort consisted of 4151 genetically determined European male individuals who had been recruited to the cancer arm of the 100KGP, excluding those with kidney, bladder, prostate, or childhood malignancy. These individuals had not been used in the discovery analysis.

Only individuals with high-confidence genotype calls were included in the analysis. Allele counts at each variant were compared between cases and controls using a two-sided Cochran-Armitage trend test. A Bonferroni-corrected p<0.0125 (0.05/4) was used to adjust for the number of loci tested. Power to detect or refute association at each locus was calculated as >0.9.

## Power

Statistical power for single-variant association under an additive model for the discovery and replication cohorts was calculated using the R package genpwr (*Moore et al., 2021*). *Figure 2—figure supplement 2* shows the power calculations for the diverse ancestry seqGWAS at varying allele frequencies and odds ratios.

## Bayesian fine mapping

We applied PAINTOR (version 3.1; *Kichaev et al., 2020*), a statistical fine-mapping method which uses an empirical Bayes prior to integrate functional annotation data, LD patterns, and strength of association to estimate the PP of a variant being causal. Variants at each genome-wide significant locus with p<0.05 were extracted. Z-scores were calculated as effect size (β) divided by standard error. LD matrices of pairwise correlation coefficients were derived using EUR 1000 Genomes (Phase 3) (*Auton et al., 2015*) imputed data as a reference, excluding variants with ambiguous alleles (A/T or G/C). Each locus was intersected with the following functional annotations downloaded using UCSC Table Browser (*Kent et al., 2002*): GENCODE (*Frankish et al., 2019*) (version 29) transcripts (wgEncodeGencodeBasicV29, updated 2019-02-15), PhastCons (*Siepel et al., 2005*) (phastConsElements100way, updated 2015-05-08), ENCODE (*Moore et al., 2020*) cCREs (encodeCcreCombined, updated 2020-05-20), TF-binding clusters (encRegTfbsClustered, updated 2019-05-16), DNase I hypersensitivity clusters (wgEncodeRegDnaseClustered, updated 2019-01-08) and H1 Human embryonic stem cell Hi-C data (h1hescInsitu from *Krietenstein et al., 2020*). A total of 351 variants at 12q24.21 and 166 variants at 6p21.1 were analysed under the assumption of one causal variant per locus.

## Functional annotation

To explore the functional relevance of the prioritized variants, we used FUMA (version 1.3.6a) (*Watanabe et al., 2017*) to annotate the genome-wide significant loci. This web-based tool integrates functional gene consequences from ANNOVAR (*Wang et al., 2010*), CADD (*Rentzsch et al., 2019*) scores to predict deleteriousness, RegulomeDB score to indicate potential regulatory function (*Boyle et al., 2016*) and 15-core chromatin state (predicted by ChromHMM for 127 tissue/cell types; *Ernst and Kellis, 2012*) representing accessibility of genomic regions. Positional mapping (where a variant is physically located within a 10 kb window of a gene), GTEx (version 8) eQTL data (*Consortium, 2020*) (using *cis*-eQTLs to map variants to genes up to 1 Mb apart) and Hi-C data to detect long-range 3D chromatin interactions is used to prioritize genes that are likely to be affected by variants of interest. Single-variant seqGWAS summary statistics were used as input with genomic positions converted to GRCh37 using the UCSC liftOver tool (*Kent et al., 2002*).

In addition, we intersected prioritized variants with the following epigenomic datasets from male H1-BMP4 derived mesendoderm cultured cells generated by the ENCODE Project (*Moore et al., 2020*) and Roadmap Epigenomics (*Kundaje et al., 2015*) Consortia using the UCSC Genome Browser (*Kent et al., 2002*): ENCFF918FRW_ENCFF748XLQ_ENCFF313DOD (cCREs, GRCh38); ENCFF918FRW_ENCFF748XLQ_ENCFF313DOD_ENCFF313DOD (H3K27ac ChIP-seq, GRCh38); ENCFF918FRW_ENCFF748XLQ_ENCFF313DOD_ENCFF748XLQ (H3K4me3 ChIP-seq, GRCh38); ENCFF918FRW_ENCFF748XLQ_ENCFF313DOD_ENCFF918FRW (DNase-seq, GRCh38); E004 H1 BMP4 Derived Mesendoderm Cultured Cells ImputedHMM (hg19). Hi-C interactions from H1 mesendoderm cells (*Kundaje et al., 2015*) and TADs were visualized with the 3D Interaction Viewer and Database (http://3div.kr).

## Gene and gene set analysis

MAGMA (version 1.6; *de Leeuw et al., 2015*) was used to test the joint association of all variants within a particular gene or gene set using the single-variant seqGWAS summary statistics. Aggregation of variants increases power to detect multiple weaker associations and can test for association with specific biological or functional pathways. MAGMA uses a multiple regression approach

to account for LD between variants, using a reference panel derived from 10,000 Europeans in the UK Biobank (release 2b). Variants from the seqGWAS were assigned to 18,757 protein coding genes (Ensembl build 85) with genome-wide significance defined as $p=2.67 \times 10^{-6}$ (0.05/18,757). Competitive gene set analysis was then performed for 5497 curated gene sets and 9986 Gene Ontology (GO) terms from MsigDB (version 7.0; *Liberzon et al., 2015*) using the results of the gene analysis. Competitive analysis tests whether the joint association of genes in a gene set is stronger than a randomly selected set of similarly sized genes. Bonferroni correction was applied for the total number of tested gene sets ($p=0.05/15,483=3.23 \times 10^{-6}$).

## Identification of TFBS

The JASPAR 2020 (*Fornes et al., 2020*) CORE collection track (UCSC Genome Browser [*Kent et al., 2002*], updated 2019-10-13) was utilized to identify significant ($p<10^{-4}$) predicted TFBS that might intersect with the lead variants. The JASPAR database consists of manually curated, non-redundant, experimentally defined TF-binding profiles for 746 vertebrates, of which 637 are associated with human TF with known DNA-binding profiles. Sequence logos based on position weight matrices of the DNA-binding motifs were downloaded from JASPAR 2020 (*Fornes et al., 2020*).

## GWAS and PheWAS associations

The NHGRI-EBI GWAS Catalog (*MacArthur et al., 2017*) and PheWAS data from the UK Biobank (https://pheweb.org/UKB-SAIGE) were interrogated to determine known associations of the lead variants. Summary statistics were downloaded from the NHGRI-EBI GWAS Catalog (*MacArthur et al., 2017*) for study GCST002890 (*Berndt et al., 2015*) on 2021-03-17. PheWAS statistics were generated using imputed data from White British participants in the UK Biobank using SAIGE, adjusting for genetic relatedness, sex, birth year, and the first four principal components.

## Immunohistochemistry

Human embryonic tissues, collected after maternal consent and ethical approval (REC18/NE/0290), were sourced from the Medical Research Council and Wellcome Trust Human Developmental Biology Resource (https://www.hdbr.org/). Tissue sections were immunostained, as we described previously (*Kolvenbach et al., 2019*). Sections were immunostained with the following primary antibodies: TBX5 (https://www.abcam.com/tbx5-antibody-ab223760.html) raised in rabbit; PTK7 (https://www.thermofisher.com/antibody/product/PTK7-Antibody-Polyclonal/PA5-82070) raised in rabbit; and uroplakin 1B (https://www.abcam.com/uroplakin-ibupib-antibody-upk1b3081-ab263454.html) raised in mouse. Primary antibodies were detected with appropriate second antibodies and signals generated with a peroxidase-based system.

## Aggregate rare coding variant analysis

Single-variant association testing is underpowered when variants are rare and a collapsing approach which aggregates variants by gene can be adopted to boost power. We extracted coding SNVs and indels with MAF <0.1% in gnomAD (*Karczewski et al., 2020*), annotated with one of the following: missense, in-frame insertion, in-frame deletion, start loss, stop gain, frameshift, splice donor, or splice acceptor. Variants were further filtered by CADD score (version 1.5; *Rentzsch et al., 2019*) using a threshold of ≥20 corresponding to the top 1% of all predicted deleterious variants in the genome. Variants meeting the following quality control filters were retained: MAC ≤20, median site-wide depth in non-missing samples >20 and median GQ ≥30. Sample-level QC metrics for each site were set to minimum depth per sample of 10, minimum GQ per sample of 20, and ABratio p-value >0.001. Variants with significantly different missingness between cases and controls ($p<10^{-5}$) or >5% missingness overall were excluded. We employed SAIGE-GENE (version 0.42.1; *Zhou et al., 2016b*) to ascertain whether rare coding variation was enriched in cases on a per-gene basis exome-wide. Like SAIGE, SAIGE-GENE utilizes a GLMM to correct for population stratification and cryptic relatedness as well as a saddlepoint approximation and efficient resampling adjustment to account for the inflated type 1 error rates seen with unbalanced case-control ratios. It combines single-variant score statistics and their covariance estimate to perform SKAT-O (*Lee et al., 2012*) gene-based association testing, upweighting rarer variants using the β(1,25) weights option. SKAT-O (*Lee et al., 2012*) is a combination of a traditional burden and variance-component test and provides robust power when the

underlying genetic architecture is unknown. Sex and the top 10 principal components were included as fixed effects when fitting the null model. Chromosome X analysis was performed in males only. A Bonferroni adjusted p-value of $2.58 \times 10^{-6}$ (0.05/19,364 genes) was used to determine the exome-wide significance threshold.

### SV analysis

SVs (>50 bp) that intersect by a minimum of 1 bp with (a) at least one exon ([GENCODE version 29 [*Frankish et al., 2019*]]) or (b) an ENCODE (*Moore et al., 2020*) cCRE were extracted using BEDTools (version 2.27.1) (*Quinlan and Hall, 2010*). Variants were retained if they fulfilled the following quality filters: Q-score ≥Q10 (CANVAS [*Roller et al., 2016*]) or QUAL ≥20, GQ ≥15, and MaxMQ0Frac <0.4 (MANTA [*Chen et al., 2016a*]). Variants without paired read support, inconsistent ploidy, or depth >×3 the mean chromosome depth near breakends were excluded.

ENCODE (*Moore et al., 2020*) cCREs are 150–350 bp consensus sites of chromatin accessibility (DNase hypersensitivity sites) with high H3K4me3, high H3K27ac, and/or high CTCF signal in at least one biosample. A list of 926,535 cCREs encoded by 7.9% of the human genome was downloaded from UCSC Table Browser using the encodeCcreCombined track (updated 2020-05-20). This includes ~668,000 dELS elements, ~142,000 pELS elements, ~57,000 CTCF-only elements, ~35,000 PLS, and ~26,000 DNase-H3K4me3 elements (promoter-like signals >200 bp from a transcription start site).

Variants were separated and filtered by SV type (deletion, duplication, CNV, inversion); those with a minimum 70% reciprocal overlap with common SVs from (a) dbVar (*MacDonald et al., 2014*) or (b) 12,234 cancer patients from the 100KGP were removed. The dbVar NCBI-curated dataset of SVs (nstd186) contains variant calls from studies with at least 100 samples and AF >1% in at least one population, including gnomAD (*Karczewski et al., 2020*), 1000 Genomes (Phase 3; *Auton et al., 2015*) and DECIPHER (*Cooper et al., 2011*). To create a dataset of common SVs from the 100KGP cancer cohort, variants were merged using SURVIVOR (version 1.0.7; *Jeffares et al., 2017*), allowing a maximum distance of 300 bp between pairwise breakpoints, and those with AF >0.1% retained. After removal of overlapping common variants, SVs in the case-control cohort were filtered to keep those with AF <0.1% and aggregated across 19,907 autosomal protein-coding genes and five cCRE types. Exome-wide gene-based and genome-wide cCRE-based burden analysis was carried out using custom R scripts (*Source code 2*). The burden of rare autosomal SVs in cases and controls was enumerated by comparing the number of individuals with ≥1 SV using a two-sided Fisher's exact test. The Wilcoxon Mann-Whitney test was used to compare median SV size. Bonferroni adjustment for the number of genes (p=0.05/19,907=$2.5 \times 10^{-6}$) and cCRE/SV combinations (p=0.05/20=$2.5 \times 10^{-3}$) tested was applied.

## Acknowledgements

This research was made possible through access to the data and findings generated by the 100KGP. The 100KGP is managed by Genomics England Limited (a wholly owned company of the Department of Health and Social Care). The 100KGP is funded by the National Institute for Health Research and NHS England. The Wellcome Trust, Cancer Research UK, and the Medical Research Council have also funded research infrastructure. The 100KGP uses data provided by patients and collected by the National Health Service as part of their care and support. The authors gratefully acknowledge the participation of the patients and their families recruited to the 100KGP. The authors also gratefully acknowledge the participation of the patients and their families in the replication study, the majority of whom were recruited via the CaRE for LUTO Study.

## Additional information

#### Group author details

**Genomics England Research Consortium**
**JC Ambrose; P Arumugam; R Bevers; M Bleda; F Boardman-Pretty; CR Boustred; H Brittain; MA Brown; MJ Caulfield; GC Chan; A Giess; JN Griffin; A Hamblin; S Henderson; TJP Hubbard; R**

Jackson; LJ Jones; D Kasperaviciute; M Kayikci; A Kousathanas; L Lahnstein; A Lakey; SEA Leigh; IUS Leong; FJ Lopez; F Maleady-Crowe; M McEntagart; F Minneci; J Mitchell; L Moutsianas; M Mueller; N Murugaesu; AC Need; P O'Donovan; CA Odhams; C Patch; D Perez-Gil; MB Pereira; J Pullinger; T Rahim; A Rendon; T Rogers; K Savage; K Sawant; RH Scott; A Siddiq; A Sieghart; SC Smith; A Sosinsky; A Stuckey; M Tanguy; AL Taylor Tavares; ERA Thomas; SR Thompson; A Tucci; MJ Welland; E Williams; K Witkowska; SM Wood; M Zarowiecki

## Competing interests
Genomics England Research Consortium: The other authors declare that no competing interests exist.

## Funding

| Funder | Grant reference number | Author |
| --- | --- | --- |
| Kidney Research UK | TF_004_20161125 | Melanie MY Chan |
| Medical Research Council | MR/S021329/1 | Omid Sadeghi-Alavijeh |
| Medical Research Council | MR/T016809/1 | Adrian S Woolf |
| St Peter's Trust for Kidney Bladder & Prostate Research | | Daniel P Gale |
| National Institute for Health and Care Research | | Adam P Levine |
| Kidney Research UK | Paed_RP_002_20190925 | Glenda M Beaman |
| BONFOR-Gerok Grant | | Alina C Hilger |

The funders had no role in study design, data collection and interpretation, or the decision to submit the work for publication. For the purpose of Open Access, the authors have applied a CC BY public copyright license to any Author Accepted Manuscript version arising from this submission.

## Author contributions
Melanie MY Chan, Conceptualization, Data curation, Formal analysis, Funding acquisition, Validation, Investigation, Visualization, Methodology, Writing – original draft, Project administration; Omid Sadeghi-Alavijeh, Data curation, Formal analysis, Methodology, Project administration, Writing – review and editing; Filipa M Lopes, Validation, Investigation, Visualization; Alina C Hilger, Validation, Writing – review and editing; Horia C Stanescu, Supervision; Catalin D Voinescu, Software; Glenda M Beaman, William G Newman, Marcin Zaniew, Stefanie Weber, Validation; Yee Mang Ho, Investigation; John O Connolly, Dan Wood, Resources, Project administration; Carlo Maj, Data curation, Formal analysis; Alexander Stuckey, Athanasios Kousathanas, Software, Methodology; Genomics England Research Consortium, Data curation, Resources; Robert Kleta, Supervision, Writing – review and editing; Adrian S Woolf, Supervision, Validation, Methodology, Writing – review and editing; Detlef Bockenhauer, Supervision, Methodology, Writing – review and editing; Adam P Levine, Conceptualization, Software, Formal analysis, Supervision, Investigation, Methodology, Writing – review and editing; Daniel P Gale, Conceptualization, Supervision, Funding acquisition, Methodology, Writing – original draft, Project administration

## Author ORCIDs
Melanie MY Chan http://orcid.org/0000-0003-1968-1734
Catalin D Voinescu http://orcid.org/0000-0002-3636-8689
Adrian S Woolf http://orcid.org/0000-0001-5541-1358
Daniel P Gale http://orcid.org/0000-0002-9170-1579

## Ethics
Human subjects: Ethical approval for the 100,000 Genomes Project was granted by the Research Ethics Committee for East of England - Cambridge South (REC Ref 14/EE/1112). Written informed consent was obtained from all participants and/or their guardians. Human embryonic tissues, collected after maternal consent and ethical approval (REC18/NE/0290), were sourced from the Medical Research Council and Wellcome Trust Human Developmental Biology Resource (https://www.hdbr.org/).

**Decision letter and Author response**
Decision letter https://doi.org/10.7554/eLife.74777.sa1
Author response https://doi.org/10.7554/eLife.74777.sa2

# Additional files

## Supplementary files
- Transparent reporting form
- Source code 1. Ancestry matching.
- Source code 2. SV burden code.

## Data availability

All genetic and phenotypic data from the 100,000 Genomes Project and can be accessed by application to Genomics England Ltd (https://www.genomicsengland.co.uk/about-gecip/joining-research-community/). Access is free for academic research institutions and universities as well as public and private healthcare organisations that undertake significant research activity. This dataset includes de-identified, linked information for each participant including genome sequence data, variant call files, phenotype/clinical data and Hospital Episode Statistics (HES) with access gained through a secure Research Environment. No sequencing or identifiable personal data is available for download. The full GWAS summary statistics have been uploaded to the NHGRI-EBI GWAS Catalog (http://ftp.ebi.ac.uk/pub/databases/gwas/summary_statistics/GCST90134001-GCST90135000/GCST90134254). Source data files have been provided for Figures 2, 6, 9 and 10 containing the numerical data used to generate figures. Code for SAIGE and SAIGE-GENE can be found at https://github.com/weizhouU-MICH/SAIGE, (copy archived at swh:1:rev:a41727267cb5f843a4446e4d4809cafc72687a5d). Code for PAINTOR is available at https://github.com/gkichaev/PAINTOR_V3.0, (copy archived at swh:1:rev:ff-ce82a7114eeb98220db39ba5d6b69e5a419a9e) Functional annotation and MAGMA gene and gene-set analysis were performed using the web-based platform FUMA (https://fuma.ctglab.nl). Custom R Code for the case-control ancestry-matching and structural variant burden analysis have been uploaded as Source code 1 (ancestry matching) and Source code 2 (SV burden testing).

The following dataset was generated:

| Author(s) | Year | Dataset title | Dataset URL | Database and Identifier |
|---|---|---|---|---|
| Chan MMY, Sadeghi-Alavijeh O, Genomics England Research Consortium, Levine AP, Bockenhauer D, Gale DP | 2022 | PUV Summary Statistics | http://ftp.ebi.ac.uk/pub/databases/gwas/summary_statistics/GCST90134001-GCST90135000/GCST90134254 | GWAS Catalog, GCST90134254 |

The following previously published datasets were used:

| Author(s) | Year | Dataset title | Dataset URL | Database and Identifier |
|---|---|---|---|---|
| Dixon JR, Jung I, Selvaraj S | 2015 | Chromatin architecture reorganization during stem cell differentiation | https://www.ncbi.nlm.nih.gov/geo/query/acc.cgi?acc=GSE52457 | NCBI Gene Expression Omnibus, GSE52457 |
| Buniello A, MacArthur JAL, Cerezo M | 2019 | The NHGRI-EBI GWAS Catalog of published genome-wide association studies, targeted arrays and summary statistics 2019 | https://www.ebi.ac.uk/gwas/studies/GCST002890 | GWAS Catalog, GCST002890 |

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

## Appendix 1

### Pathogenic/likely pathogenic variants identified in PUV cases

One individual was heterozygous for a pathogenic 1.5 Mb 17q12 deletion (affecting HNF1B) which has previously been associated with autosomal-dominant renal cysts and diabetes syndrome (RCAD, MIM 137920). Review of available clinical information revealed the presence of renal cysts, hypomagnesemia, and hypokalemia, which is consistent with HNF1B-related disease. There was no reported family history which is often seen since ~50% of whole gene deletions are de novo and there is variable expressivity. Identification of this variant was reported back to the clinical team for validation and to inform surveillance for additional manifestations (e.g., diabetes, hyperuricemia, autism) and screening of family members. Review of hospital records confirmed that this individual had undergone endoscopic valve ablation for PUV as an infant and therefore had two separate diagnoses with the 17q12 deletion not determined as causal for PUV.

Another individual with a diagnosis of PUV, ocular hypertension, neurodevelopmental delay, and autistic behavior was found to be heterozygous for a likely pathogenic missense variant FOXC1:c.379C>T; p.(Arg127Cys). This variant has been previously reported in individuals with pediatric glaucoma (*Khalil et al., 2017*) and heterozygous FOXC1 variants have been identified in individuals with syndromic CAKUT, one of whom had PUV (*Wu et al., 2020*). After multi-disciplinary review of the available evidence, this variant was not thought to fully explain the patient's phenotype.

