## [Editor Report]

Prior work has linked posterior urethral valves (PUV), a common cause of end stage renal disease in children, with chromosomal abnormalities and rare copy number variants, but the genetic causes of PUV remain incompletely defined. In this study, the authors have used a diverse ancestry whole-genome sequencing association study to identify two novel genes, *TBX5* and *PTK7*, and an enrichment of rare duplications and inversions affecting candidate *cis*-regulatory elements as possible causes of this condition. These findings represent the first robust genetic associations of PUV, provide the foundation for developing a mechanistic understanding of the cause of this condition, and demonstrate how a diverse ancestry seqGWAS can be used for disease locus discovery in a rare disease.

---

## [Decision Letter]

**Decision letter after peer review:**

Thank you for submitting your article "Diverse ancestry whole-genome sequencing association study identifies *TBX5* and *PTK7* as susceptibility genes for posterior urethral valves" for consideration by *eLife*. Your article has been reviewed by 3 peer reviewers, one of whom is a member of our Board of Reviewing Editors, and the evaluation has been overseen by Martin Pollak as the Senior Editor. The following individual involved in review of your submission has agreed to reveal their identity: Jason Flannick (Reviewer #3).

Essential revisions:

As you can see from the detailed comments below, the reviewers were generally quite positive about the study but did have several issues they wanted addressed before acceptance. Through our consultation process, we came to a consensus that a satisfactory response to any one of the three items would be acceptable:

1. The authors provide more direct biological evidence showing that variants associated with PUV affect TBX5 or PTK7 expression while other nearby variants not associated with PUV do not; or that altering the expression of either gene results in GU abnormalities.

2. The authors provide stronger evidence in support of the PTK7 locus with more stringent analysis and more robust replication.

3. If it is not practically possible to provide stronger evidence in support of the PTK7 locus in a timely way, the authors should otherwise address the major concerns raised by Reviewer 2.

The reviewers have also made a number of other non-essential recommendations that they suggest you consider in your resubmission.

*Reviewer #1 (Recommendations for the authors):*

The genetic studies are well-done and provide a good basis for discovery of novel processes causing this important condition. The identification of TBX5 as a potential cause of PUV is particularly interesting given that other TBX genes have also been associated with CAKUT (which includes PUV). However, the study's conclusions would be greatly strengthened with experimental validation. Perhaps the authors could do CRISPR to the mesodermal cell line, introducing the variant associated with protection from PUV, and then show how this affects TBX5 and PTK7 expression. Or they could use the approach of Verbitsky et al. and characterize mouse models to show that variants in the putative genes result in a GU phenotype in males.

One trivial observation: the authors should explain the significance of the green arrow in Figure 2B that crosses the DNAse line. It wasn't obvious from either the text or the figure legend what this arrow was pointing to or indicating.

*Reviewer #2 (Recommendations for the authors):*

1. The sample size is small for genetic discovery and more quantitative discussion of power limitations should be included in the discussion of limitations, mainly to help the readers understand what variant effects are detectable in the discovery phase. The power plot clearly demonstrates that the present design has limited power for variants with MAF<0.01, and this should be discussed especially in the context of the PTK7 locus.

2. PheWAS findings from the UK Biobank for TBX5 locus are interesting, providing independent validation of the role of this gene in urogenital development. It would be helpful to include a PheWAS plot and mention if there are any non-urogenital associations (e.g. cardiac phenotypes, limb defects, other).

3. Gene-based burden tests were largely negative but it would be helpful to see QQ plots and the assessment of genomic inflation (with sample size correction).

4. Page 12 Line 182 "the two variants were not in strong linkage disequilibrium (LD; r2 183 = 0.54)." Please state ancestral population used whenever providing LD estimates.

5. Some of the discussion language comes across as too strong given the small size of the case cohort. For example, I would suggest changing more general statements such as "monogenic causes of PUV are not a common feature" to be more specific by adding "known monogenic causes" and "in our cohort". In my opinion, limited sample size of this study precludes these more general conclusions. There may be unknown Monogenic causes in the cohorts that the authors are unable to identify.

*Reviewer #3 (Recommendations for the authors):*

This paper is well-written, and the analysis approach is very strong. Overall I have very few objections to anything in the manuscript. Here are a few suggestions that the authors could consider that I think would further increase the impact of this work:

1. I found the background on what is known about the genetic architecture of PUV to be a bit cursory. I think the introduction indicates that a monogenic eitiology of PUV has not been identified. Does this mean that no genes have been implicated in the disease? Or have some cases been solved but others remain to be solved? What is the inheritance pattern? Is it Mendelian or complex? A clearer description would better motivate the GWAS as the appropriate study design.

2. I felt that the motivation for replication study was a little unclear. If the cases were matched to controls in the 100KGP, why weren't they included from the start? Were the matched controls intentionally held out of the discovery analysis? Was it just chance that the best matches for the replication controls happened not to be included in the discovery study?

3. I didn't find a discussion as to how much of the genetic basis of PUV were explained by the new variants. This is related to point #1 above. Are there now "solved" cases? Or are these just risk factors in the population? How much heritability is explained by them? How big of an impact do they have in understanding the basis of PUV?

4. The conclusion that monogenic causes of PUV are rare does not follow in my opinion from the lack of significant burden test results. Monogenic causes could exist but not produce significant associations due to power issues or due to the inclusion of too many benign variants. Conversely, burden test results may not reveal monogenic causes but instead just moderate effect variants (most burden test results to date are for common diseases). I think these conclusions need to be more carefully worded, or their logic needs to be more clearly spelled out.

5. I didn't understand the impact of the genome-wide SV analyses. I think I follow that previous studies have shown a role for SVs in disease. But global enrichments, or even enrichments across all cREs, seem to tell us so little about disease. I think the model for this analysis, and what it tells us about disease beyond a broad enrichment, needs to be more clearly spelled out. At minimum, how much of the genetic basis of PUV is explained by these variants? And, ideally, what implications does it have for how we might diagnose or treat PUV?

---

## [Author Response]

Reviewer #1 (Recommendations for the authors):The genetic studies are well-done and provide a good basis for discovery of novel processes causing this important condition. The identification of TBX5 as a potential cause of PUV is particularly interesting given that other TBX genes have also been associated with CAKUT (which includes PUV). However, the study's conclusions would be greatly strengthened with experimental validation.

We would like to thank the reviewer for their comments regarding the robustness of the genetic associations and bioinformatic approaches used. Whilst we appreciate that this manuscript does not include in vitro or in vivo functional experiments, we have shown a clear statistically significant difference between individuals with PUV and an ancestry-matched control population, which we have replicated in an independent cohort. The strength of this study lies in its statistical robustness and whilst we agree that functional experiments will certainly be necessary to better understand the biological mechanisms underpinning these findings, these will be the focus of subsequent work. We have clearly stated this as a limitation of the study on page 23, lines 524-527.

Perhaps the authors could do CRISPR to the mesodermal cell line, introducing the variant associated with protection from PUV, and then show how this affects TBX5 and PTK7 expression. Or they could use the approach of Verbitsky et al. and characterize mouse models to show that variants in the putative genes result in a GU phenotype in males.

Thank you for these very useful suggestions. We envisage that publication of these statistically robust genetic findings will catalyze further mechanistic studies such as those mentioned by the reviewer above by both us and other groups in the field, but such experiments would likely be the subject of a subsequent manuscript. However, we would like to emphasise again, that ours is an association study and we therefore do not describe a Mendelian disorder, in which rare and highly penetrant variants cause a specific disorder. Rather, the statistically significant enrichment in the cases compared to controls of the variants in our study imply that these variants provide a susceptibility for developing PUV but are not causative on their own. This clearly complicates mechanistic studies, which is why we believe, that they will constitute subsequent work.

With regards to evidence in support of these genes causing a genitourinary phenotype, we have added additional phenotypic detail from the published literature to the discussion (page 19-20, line 445-449) that show that mesoderm-specific conditional deletion of *Ptk7* in mice results in genitourinary abnormalities (Xu et al. 2016; Xu, Santos, and Hinton 2018). Furthermore, the PheWAS data detailed on page 13, lines 290-300 provide independent validation of a role for *TBX5* in lower urinary tract phenotypes in humans.

One trivial observation: the authors should explain the significance of the green arrow in Figure 2B that crosses the DNAse line. It wasn't obvious from either the text or the figure legend what this arrow was pointing to or indicating.

In Figure 3, there is a green box signifying a *cis*-regulatory element underneath the green DNase signal plot which we believe the reviewer may have mistaken for an arrow. This may be a formatting issue and we will confirm the final figure is clear before publication.

Reviewer #2 (Recommendations for the authors):1. The sample size is small for genetic discovery and more quantitative discussion of power limitations should be included in the discussion of limitations, mainly to help the readers understand what variant effects are detectable in the discovery phase. The power plot clearly demonstrates that the present design has limited power for variants with MAF<0.01, and this should be discussed especially in the context of the PTK7 locus.

A more detailed comment on the power limitations of the study has been added to the discussion of limitations as suggested (page 22, line 514-517).

2. PheWAS findings from the UK Biobank for TBX5 locus are interesting, providing independent validation of the role of this gene in urogenital development. It would be helpful to include a PheWAS plot and mention if there are any non-urogenital associations (e.g. cardiac phenotypes, limb defects, other).

This has been added as Figure 7. No non-urogenital associations were detected with this variant which has been added to the results (page 13, line 297-300).

3. Gene-based burden tests were largely negative but it would be helpful to see QQ plots and the assessment of genomic inflation (with sample size correction).

The Q-Q plot for the rare variant gene burden analysis has been added as Figure 9 – supplemental figure 1. When the case:control ratio is very unbalanced (as in this analysis), the allele counts in cases for rare variants can be very low and the association tests lack power resulting in a deflation at the bottom left corner of the QQ plot. For this reason, the genomic inflation factor (λ) can be unreliable and we suggest it would not add any useful additional information to the interpretation of the results in this context, especially in the absence of any significant results.

4. Page 12 Line 182 "the two variants were not in strong linkage disequilibrium (LD; r2 183 = 0.54)." Please state ancestral population used whenever providing LD estimates.

This has been added to the manuscript on page 9, lines 193-194.

5. Some of the discussion language comes across as too strong given the small size of the case cohort. For example, I would suggest changing more general statements such as "monogenic causes of PUV are not a common feature" to be more specific by adding "known monogenic causes" and "in our cohort". In my opinion, limited sample size of this study precludes these more general conclusions. There may be unknown Monogenic causes in the cohorts that the authors are unable to identify.

We appreciate the reviewer’s point, although would reply that we have simply stated that monogenic causes of PUV are not a ‘common feature’ and there are indeed no proven Mendelian monogenic causes of non-syndromic PUV that have been identified. We have however moderated some of the language as suggested in the Results (page 14, line 318 and line 321, and page 15, lines 336-338) and Discussion (page 17, line 382-383; page 22, lines 514-517).

Reviewer #3 (Recommendations for the authors):This paper is well-written, and the analysis approach is very strong. Overall I have very few objections to anything in the manuscript. Here are a few suggestions that the authors could consider that I think would further increase the impact of this work:1. I found the background on what is known about the genetic architecture of PUV to be a bit cursory. I think the introduction indicates that a monogenic eitiology of PUV has not been identified. Does this mean that no genes have been implicated in the disease? Or have some cases been solved but others remain to be solved? What is the inheritance pattern? Is it Mendelian or complex? A clearer description would better motivate the GWAS as the appropriate study design.

Many thanks for highlighting this, we have expanded and clarified the background section as suggested (page 4, lines 74-93).

2. I felt that the motivation for replication study was a little unclear. If the cases were matched to controls in the 100KGP, why weren't they included from the start? Were the matched controls intentionally held out of the discovery analysis? Was it just chance that the best matches for the replication controls happened not to be included in the discovery study?

We have clarified this in the methods section (Page 29, lines 674-692). The 100KGP consists of two main cohorts: cancer patients and rare disease patients and their families. The control population used for the discovery cohort were the unaffected relatives of participants recruited to the 100KGP with non-renal rare diseases. Given this was > 20,000 individuals, there was no appreciable increase in power by addition of more controls from the cancer arm of the 100KGP. For the replication study, we used the ~10,000 additional EUR patients who had been recruited to the 100KGP with non-renal or urinary tract adult-onset cancer (but had not been used in the discovery stage).

3. I didn't find a discussion as to how much of the genetic basis of PUV were explained by the new variants. This is related to point #1 above. Are there now "solved" cases? Or are these just risk factors in the population? How much heritability is explained by them? How big of an impact do they have in understanding the basis of PUV?

We thank the reviewer for pointing this out and have added a paragraph regarding what these results indicate for the genetic basis of PUV to the discussion (page 21, line 494--502). Unfortunately, the cohort size is too small to produce a reliable estimate of SNP-heritability: we did use GCTA’s GREML-LDMS approach to estimate the *observed* narrow-sense heritability attributable to variants with MAF > 0.1% to be 0.31 (SE 0.05) in the European subset of the cohort. However, with the small sample size and low disease prevalence this cannot be transformed reliably onto the liability scale which would allow comparison with other cohorts. For this reason, we made the decision not to include these data in the manuscript.

4. The conclusion that monogenic causes of PUV are rare does not follow in my opinion from the lack of significant burden test results. Monogenic causes could exist but not produce significant associations due to power issues or due to the inclusion of too many benign variants. Conversely, burden test results may not reveal monogenic causes but instead just moderate effect variants (most burden test results to date are for common diseases). I think these conclusions need to be more carefully worded, or their logic needs to be more clearly spelled out.

We appreciate the reviewer’s comments and have moderated our discussion of the rare variant burden analysis in the Results section and discussion.

5. I didn't understand the impact of the genome-wide SV analyses. I think I follow that previous studies have shown a role for SVs in disease. But global enrichments, or even enrichments across all cREs, seem to tell us so little about disease. I think the model for this analysis, and what it tells us about disease beyond a broad enrichment, needs to be more clearly spelled out. At minimum, how much of the genetic basis of PUV is explained by these variants? And, ideally, what implications does it have for how we might diagnose or treat PUV?

Please see response to Reviewer 2 – major point 3: we have stressed the hypothesis generating and exploratory nature of these analyses in the discussion (page 21, lines 491-493) and suggest that ‘ non-specific perturbation of long-range regulatory networks or TADs could manifest as PUV, perhaps due to sensitivity of integration of the mesonephric duct into the posterior urethra to even minor abnormalities of gene expression’. Given the exploratory nature of these analyses we think it would be premature to draw any more concrete conclusions regarding heritability, diagnosis, and treatment.

References

Gimelli, Stefania, Gianluca Caridi, Silvana Beri, Kyle McCracken, Renata Bocciardi, Paola Zordan, Monica Dagnino, et al. 2010. “Mutations in SOX17 Are Associated with Congenital Anomalies of the Kidney and the Urinary Tract.” Human Mutation 31 (12): 1352–59.

Krishnan, Anand, Antonio de Souza, Rama Konijeti, and Laurence S. Baskin. 2006. “The Anatomy and Embryology of Posterior Urethral Valves.” The Journal of Urology 175 (4): 1214–20.

Ober, Carole, Dagan A. Loisel, and Yoav Gilad. 2008. “Sex-Specific Genetic Architecture of Human Disease.” Nature Reviews. Genetics 9 (12): 911–22.

Ruf, Rainer G., Pin-Xian Xu, Derek Silvius, Edgar A. Otto, Frank Beekmann, Ulla T. Muerb, Shrawan Kumar, et al. 2004. “SIX1 Mutations Cause Branchio-Oto-Renal Syndrome by Disruption of EYA1-SIX1-DNA Complexes.” Proceedings of the National Academy of Sciences of the United States of America 101 (21): 8090–95.

Sanna-Cherchi, Simone, Rosemary V. Sampogna, Natalia Papeta, Katelyn E. Burgess, Shannon N. Nees, Brittany J. Perry, Murim Choi, et al. 2013. “Mutations in DSTYK and Dominant Urinary Tract Malformations.” The New England Journal of Medicine 369 (7): 621–29.

Shin, Jimin, and Chaeyoung Lee. 2015. “A Mixed Model Reduces Spurious Genetic Associations Produced by Population Stratification in Genome-Wide Association Studies.” Genomics 105 (4): 191–96.

Xu, Bingfang, Sérgio A. A. Santos, and Barry T. Hinton. 2018. “Protein Tyrosine Kinase 7 Regulates Extracellular Matrix Integrity and Mesenchymal Intracellular RAC1 and Myosin II Activities during Wolffian Duct Morphogenesis.” Developmental Biology 438 (1): 33–43.

Xu, Bingfang, Angela M. Washington, Raquel Fantin Domeniconi, Ana Cláudia Ferreira Souza, Xiaowei Lu, Ann Sutherland, and Barry T. Hinton. 2016. “Protein Tyrosine Kinase 7 Is Essential for Tubular Morphogenesis of the Wolffian Duct.” Developmental Biology 412 (2): 219–33.

Yang, Jian, the GIANT Consortium, Michael N. Weedon, Shaun Purcell, Guillaume Lettre, Karol Estrada, Cristen J. Willer, et al. 2011. “Genomic Inflation Factors under Polygenic Inheritance.” European Journal of Human Genetics: EJHG 19 (7): 807–12.